# On Perfect Clustering for Gaussian Processes

**Juan A. Cuesta-Albertos**                                                    *cuestaj@unican.es*
*Departamento de Matemáticas, Estadística y Computación*
*Universidad de Cantabria, Spain*

**Subhajit Dutta**                                                                    *duttas@iitk.ac.in*
*Department of Mathematics and Statistics*
*IIT Kanpur, India*

**Reviewed on OpenReview:** *https: // openreview. net/ forum? id= igDOV2KBwM*

## Abstract

In this paper, we propose a data based transformation for infinite-dimensional Gaussian processes and derive its limit theorem. For a clustering problem using mixture models, an appropriate modification of this transformation asymptotically leads to perfect separation of the populations under rather general conditions, except the scenario in which differences between clusters depend only on the locations; in which case our procedure is useless. Theoretical properties related to label consistency are studied for the $k$-means clustering algorithm when used on this transformed data. Good empirical performance of the proposed methodology is demonstrated using simulated as well as benchmark data sets, when compared with some popular parametric and nonparametric methods for such functional data.

## 1 Introduction

Suppose that we are given two Gaussian distributions (say, GDs) $\mathbb{P}_1$ and $\mathbb{P}_2$. The Hajek and Feldman property (established independently by Hajek (1958) and Feldman (1958)) states that $\mathbb{P}_1$ and $\mathbb{P}_2$ are either equivalent, or else mutually singular. In other words, for every measurable set $A$, $\mathbb{P}_1(A) = 0$ if and only if $\mathbb{P}_2(A) = 0$, or else there exist two disjoint measurable sets $S_1$ and $S_2$ such that

$$\mathbb{P}_1(S_1) = 1, \mathbb{P}_2(S_1) = 0 \text{ and } \mathbb{P}_1(S_2) = 0, \mathbb{P}_2(S_2) = 1.$$

Mutual singularity is not very interesting in finite dimensions because it happens only when at least one of the covariance matrices is singular. However, in the functional setting, this singularity appears in non-trivial situations. To mention an example, it was shown by Rao and Varadarajan (1963) that if the covariance operators of $\mathbb{P}_1$ and $\mathbb{P}_2$, namely, $\Sigma^{\mathbb{P}_1}$ and $\Sigma^{\mathbb{P}_2}$ satisfy $\Sigma^{\mathbb{P}_2} = a\Sigma^{\mathbb{P}_1}$ for some $a \neq 1$, then $\mathbb{P}_1$ and $\mathbb{P}_2$ are mutually singular.

It is clear that the mutually singular case of the Hajek and Feldman property (say, HFp) looks very promising for classification as well as clustering (in the mixture setting) of data points. Recently, some results have appeared taking advantage of this property to propose *perfect classifiers* (see the references given below). However, the clustering problem seems to be harder and as far as we know, Delaigle et al (2019) is the only available paper with results in this area. The main drawback of the paper by Delaigle et al (2019) is that it primarily deals with location problems (see Section M of Appendix II for a detailed discussion). In this paper, we present a family of transformations on functional data which allows one to identify some mutually singular situations. The transformed data are then used to obtain *perfect clustering* in the mixture setting.

To give an overview of our main contributions, let us consider a Gaussian process (say, **Z**) defined on a bounded real interval, which without loss of generality, we identify with the unit interval $[0, 1]$. Further,

assume that its trajectories belong to the Hilbert space of square integrable functions $\mathbb{H}$, which is defined as follows:

$$\mathbb{H} : \text{set of real functions } f(t) \text{ with } t \in [0,1] \text{ such that } \int_0^1 f^2(t)dt < \infty.$$

The inner product in $\mathbb{H}$ is $\langle f, g \rangle = \int_0^1 f(t)g(t)dt$. The keystone of this paper is Theorem 2.1 in Section 2. It states that under appropriate assumptions, if $b \in \mathbb{H}$, then the limit of a sequence of scaled Mahalanobis distances between some finite-dimensional projections of $\mathbf{Z}$ and $b$ converges in probability to a non-random limit. Scaling is done using the dimension of the projection, and this convergence holds as the dimension goes to infinity. Practical interest of this result lies in the fact that the limit depends only on the distribution of $\mathbf{Z}$ (say, $\mathbb{P}_{\mathbf{Z}}$). Therefore, Theorem 2.1 allows one to identify some cases in which GDs are *mutually singular*. In such scenarios, this result allows one to obtain *perfect classification* (also see Cuesta-Albertos and Dutta (2022)) as well as *perfect clustering*. We now explain this point a bit more precisely.

Consider a probability distribution $\mathbb{P}$ such that $\mathbb{P} = \sum_{h=1}^J \pi_h \mathbb{P}_h$, where $0 < \pi_h < 1$ with $\sum_{h=1}^J \pi_h = 1$ and $\mathbb{P}_h$ are GDs on $\mathbb{H}$ for $1 \le h \le J$. Additionally, assume $\mathbb{P}$ to be *known*, but the precise values of $J$, $\pi_h$ and $\mathbb{P}_h$ for $1 \le h \le J$ are *unknown*. Thus, $\mathbb{P}_{\mathbf{Z}}$ can be understood as a two-step procedure. In the first step, we randomly select one among the $J$ possibilities with probabilities $\pi_h$ for $1 \le h \le J$. Now, if $\mathbb{P}_{h_0}$ is selected in this step, then one chooses a random function using the probability $\mathbb{P}_{h_0}$ for $h_0 \in \{1, \dots, J\}$. We will stick to this interpretation throughout the paper.

According to this mixture model, every function $\mathbf{Z} \sim \mathbb{P}$ is in fact generated from one of the $\mathbb{P}_h$'s. Using Theorem 2.1, if we project the available functions on certain subspaces of increasing dimensions, then the proposed transformation leads to different limits (depending just on the $\mathbb{P}_h$ which generated each function). Thus, if we have a set of observations (with at least one observation from each $\mathbb{P}_h$), in the limit we can identify the value of $J$ as well as the subset of the observations produced by each $\mathbb{P}_h$ without any possibility of mistake. We call this fact *perfect clustering*, which in practice means that we can estimate $J$ and the cluster assignment of all the observations correctly with a high probability when the dimension is large.

We believe that HFp should have attracted the attention of researchers in classification and clustering for functional data, the 'orthogonality case' apparently being more attractive because it would allow one to obtain *perfect classification* and *perfect clustering*. It took 50 years before the HFp was formally used in classification. To the best of our knowledge, the first paper using HFp in classification was Baillo et al (2011), where the authors derived a classification procedure using likelihood ratios. They focused on the 'equivalence case' and hence, did not obtain perfect classification. Optimal classification of Gaussian processes (say, GPs) was analyzed in Torrecilla et al (2020) from the HFp viewpoint. Further, the optimal (Bayes') classifier of equivalent GPs was derived and a procedure to obtain asymptotically perfect classification of mutually singular GPs was described as well. The results covered both homoscedastic as well as heteroscedastic cases. Additionally, Delaigle and Hall (2012) and Delaigle and Hall (2013) investigated conditions under which a perfect classification procedure for GPs was possible and developed related classifiers. The paper by Dai et al (2017) proposed a functional classifier based on ratio of density functions, which also leads to perfect classification. These papers contain no reference of the HFp. In fact, the relationship between Delaigle and Hall (2012) and the HFp was analyzed in Berrendero et al (2018), where the authors presented an expression of the optimal Bayes' rule in some classification problems for functional data. As mentioned earlier, perfect clustering has been studied only by Delaigle et al (2019).

In Rao and Varadarajan (1963) and Shepp (1966b), the authors obtained characterizations of the singularity or equivalence of Gaussian measures in functional spaces. Their results involve increasing sequences of subspaces. For equivalent GDs, the limit obtained in Rao and Varadarajan (1963) includes a term which is the exponential of an expression involving the difference of the means of $\mathbb{P}_1$ and $\mathbb{P}_2$. Curiously, the logarithm of this term is related with the expressions of our limits. Similarities between our proposal and those in Rao and Varadarajan (1963) and Shepp (1966b) end here because the other involved terms are different. Moreover, we handle Mahalanobis distances *between data points*, while these papers use Hellinger and Jeffreys functionals to measure discrepancy *between distributions*. As a consequence, the characterizations they obtain are not applicable in practice to cluster data points because they depend on the underlying distribution. Moreover, it is not straight forward to compute such functionals using data points.

### 1.1 Contributions

In this paper, we first analyze the limit of the above mentioned scaled Mahalanobis' distances by assuming the underlying parameters of the GPs to be *known* (see Section 2). We begin with a general concentration result in Theorem 2.1. Then, we propose a transformation for clustering that asymptotically yield *perfect separation* among the clusters (see Theorem 2.5). Further, this transformation can be used to find the unknown number of clusters (see Proposition 2.3). In Section 3, we *estimate* the covariance operator of the mixture distribution from data and state related asymptotic results for the proposed transformation. In Theorem 3.2, we prove uniform (on the sample points) consistency of the empirical version for the transformation associated with GP clustering. It is surprising that our GP clustering method fails to discriminate 'location only' scenarios, but yields perfect clustering for 'differences in scales' (see Remark 2.2.2). We have also compared our work both theoretically (see Section M of Appendix II) as well as numerically (see Sections 4, 5 and Section N in Appendix II) with the existing literature on perfect clustering for functional data. All proofs are deferred to Appendix I. Some additional material is presented in Appendix II, which includes a possible extension to non-Gaussian distributions (see Section K), discussion on a clustering procedure in the 'location only' case (see Section L) and theoretical comparisons of our results with those obtained in the paper by Delaigle et al (2019) (see Section M).

In summary, the main contributions of our paper are as follows:

1. Given a sample from a mixture GD, we develop a method which allows us to identify the number of members in the mixture as well as the cluster membership of the functions.

2. On the theoretical side, if we know the probability $\mathbb{P}$ (but neither $J$ nor the $\mathbb{P}_h$s) and we have a finite sample taken from $\mathbb{P}$, then the procedure allows to identify $J$ as well as the functions produced by each $\mathbb{P}_h$ for $1 \leq h \leq J$.

3. From the practical point of view, we can estimate the covariance operator of the full mixture if the sample size increases. Under appropriate conditions, our procedure allows to identify $J$ as well as the functions produced by each $\mathbb{P}_h$ with a high probability for $1 \leq h \leq J$.

4. The proposed procedure only works in those cases in which the distributions in the sample have different covariance operators and we have enough difference in the values between $L_S^h$ and $L_S^{hk}$ for $1 \leq h, k \leq J$ (see Theorem 2.2 and Remark 2.5.2).

5. Our simulations as well as analysis of real data sets show promising behaviour of the procedure when compared with several existing methods.

### 1.2 Notation

We will use the following notation. The distribution of the random process $\mathbf{Z}$ will be denoted as $\mathbb{P}_{\mathbf{Z}}$, its mean function by $\mu^{\mathbf{Z}}$ and its covariance operator (referred to simply as covariance) by $\Sigma^{\mathbf{Z}}$. We use $\Sigma^{\mathbf{Z}}(s, t)$ to denote the covariance between $\mathbf{Z}(s)$ and $\mathbf{Z}(t)$ for $s, t \in [0, 1]$. Given a square matrix $A$, $trace(A)$ denotes its trace. The usual Euclidean norm on $\mathbb{R}^d$ is denoted by $\| \cdot \|$. To simplify notation, we do not always explicitly state the dependence of the norm on the dimension $d$. Further, we will assume that all the involved random quantities are defined on a common probability space $(\Omega, \mathcal{A}, \mathbb{P})$. The notions of convergence in probability and equality in distribution are denoted by $\xrightarrow{P}$ and $\sim$, respectively.

## 2 Transformation with Known Gaussian Distributions

Let $\{V_d\}_{d \in \mathbb{N}}$ be an increasing sequence of subspaces of $\mathbb{H}$. Here, the dimension of $V_d$ is $d$. This restriction is not necessary for the development which follows as long as the dimension of $V_d$ goes to infinity with increasing $d$, but it simplifies the notation. Given the subspace $V_d$, let $\mu_d^{\mathbf{Z}}$ and $\Sigma_d^{\mathbf{Z}}$ represent the $d$-dimensional mean and the $d \times d$ covariance matrix of the projection of $\mathbf{Z}$ on $V_d$. If $\mathbf{u} \in \mathbb{H}$, we denote $\mathbf{u}_d = (u_1, \dots, u_d)^T$ to be its projection on $V_d$.

Fix $b \in \mathbb{H}$. Theorem 2.1 analyses the behaviour of the limit of squared Mahalanobis norm of the $d$-dimensional random vector $(\mathbf{Z} - b)_d$ for $d \in \mathbb{N}$. For every positive definite (p.d.) $d \times d$ matrix $A_d$, we define the map

$$D_d^{A_d}(\mathbf{u}, \mathbf{v}) = \frac{1}{d} \left\| A_d^{-1/2}(\mathbf{u} - \mathbf{v})_d \right\|^2 \text{ for } \mathbf{u}, \mathbf{v} \in \mathbb{H}. \tag{1}$$

Later, we will use this function alongwith the sequence of covariance matrices $\{\Sigma_d^{\mathbf{Z}}\}_{d \in \mathbb{N}}$.

In this section, the underlying distributions are assumed to be *known*. After stating Theorem 2.1 and some remarks related to it, we will look into an application to cluster analysis inspired from this result. We will take advantage of the fact that the limit in this theorem is not random, but it may depend on the underlying probability distribution $\mathbb{P}_{\mathbf{Z}}$.

**Theorem 2.1** *Let $\{A_d\}_{d \in \mathbb{N}}$ be a sequence of $d \times d$ symmetric, p.d. matrices and $\alpha_1^d, \ldots, \alpha_d^d$ be the eigenvalues of the matrix $S_d = (A_d)^{-1/2} \Sigma_d^{\mathbf{Z}} (A_d)^{-1/2}$ for $d \in \mathbb{N}$. We define $\alpha_d = (\alpha_1^d, \ldots, \alpha_d^d)^T$ and $\|\alpha_d\|_\infty = \max(\alpha_1^d, \ldots, \alpha_d^d)$ is the supremum norm. Let us assume that*

$$0 = \lim_{d \to \infty} \frac{\|\alpha_d\|_\infty}{d}. \tag{2}$$

*Let $b \in \mathbb{H}$ such that there exist constants $L_\mu$ and $L_S$ (finite, or not) with*

$$L_\mu = \lim_{d \to \infty} D_d^{A_d}(\mu^{\mathbf{Z}}, b) \text{ and} \tag{3}$$

$$L_S = \lim_{d \to \infty} \frac{1}{d} trace(S_d). \tag{4}$$

*Then, $D_d^{A_d}(\mathbf{Z}, b) \xrightarrow{P} L := L_\mu + L_S$ as $d \to \infty$.*

**Remark 2.1.1** A condition in Theorem 2.1 is required to ensure that no single component is extremely influential. For instance, it may happen that we take a sequence such that $\alpha_1^d = d$ and $\alpha_i^d = o(d^{-1})$ for every $2 \le i \le d$. Under this condition, a limiting value is not feasible in Theorem 2.1. However, this possibility is excluded by assumption (2).

**Remark 2.1.2** We allow both the constants in Theorem 2.1 to be infinite. When $L_S$ is finite, Lemma B.1 (see Appendix I) shows that assumption (2) follows from assumption (4).

**Remark 2.1.3** Let $\mathbf{Z}_1$ and $\mathbf{Z}_2$ be independent observations generated from the GDs $\mathbb{P}_1$ and $\mathbb{P}_2$. Thus, $\mathbf{Z}_1 - \mathbf{Z}_2$ is a GP with mean $\mu^{\mathbf{Z}_1} - \mu^{\mathbf{Z}_2}$ and covariance $\Sigma^{\mathbf{Z}_1} + \Sigma^{\mathbf{Z}_2}$. Consider the matrix $S_d = (A_d)^{-1/2}(\Sigma_d^{\mathbf{Z}_1} + \Sigma_d^{\mathbf{Z}_2})(A_d)^{-1/2}$ with $d \in \mathbb{N}$. Define $\mathbf{Z} = \mathbf{Z}_1 - \mathbf{Z}_2$ and $b = 0$ in Theorem 2.1. Then, under the assumptions of this result, the following convergence result holds:

$$D_d^{A_d}(\mathbf{Z}_1, \mathbf{Z}_2) = D_d^{A_d}(\mathbf{Z}_1 - \mathbf{Z}_2, 0) \xrightarrow{P} L := L_\mu + L_S \text{ as } d \to \infty.$$

Here, $L_\mu = \lim_{d \to \infty} D_d^{A_d}(\mu^{\mathbf{Z}_1}, \mu^{\mathbf{Z}_2})$ and $L_S$ is as defined in (4) of Theorem 2.1.

**Remark 2.1.4** In general, the fact that $V_d \subset V_{d+1}$ does not guarantee the existence of any relationship between the sets $\{\alpha_1^d, \ldots, \alpha_d^d\}$ and $\{\alpha_1^{d+1}, \ldots, \alpha_{d+1}^{d+1}\}$. However, in some cases $\{\alpha_1^d, \ldots, \alpha_d^d\} \subset \{\alpha_1^{d+1}, \ldots, \alpha_{d+1}^{d+1}\}$ holds (for instance, when $V_d$ is generated by the first $d$ eigenfunctions of $\Sigma^{\mathbf{Z}}$ in Section 2.1).

## 2.1 Application: Cluster Analysis

In this subsection, we deal with a random function $\mathbf{Z}$ whose distribution is a two component mixture distribution of the form: $\mathbb{P}_{\mathbf{Z}} = \pi_1 \mathbb{P}_1 + \pi_2 \mathbb{P}_2$, where $0 < \pi_1 < 1$ and $\pi_1 + \pi_2 = 1$. Here, $\mathbb{P}_h$ denotes the GD on $\mathbb{H}$ with $\mathbf{Z}_h \sim \mathbb{P}_h$ having mean function $\mu^{\mathbf{Z}_h}$ and covariance $\Sigma^{\mathbf{Z}_h}$ for $h = 1, 2$.

The mean function and the covariance of the mixture satisfy $\mu^{\mathbf{Z}}(t) = \pi_1 \mu^{\mathbf{Z}_1}(t) + \pi_2 \mu^{\mathbf{Z}_2}(t)$ with $t \in [0, 1]$ and

$$\Sigma^{\mathbf{Z}}(s, t) = \pi_1 \Sigma^{\mathbf{Z}_1}(s, t) + \pi_2 \Sigma^{\mathbf{Z}_2}(s, t) + \pi_1 \pi_2 [\mu^{\mathbf{Z}_1}(s) - \mu^{\mathbf{Z}_2}(s)][\mu^{\mathbf{Z}_1}(t) - \mu^{\mathbf{Z}_2}(t)] \text{ for } s, t \in [0, 1]. \tag{5}$$

Given $N$ independent and identically distributed (i.i.d.) r.v.'s $\mathbf{Z}_1, \ldots, \mathbf{Z}_N \sim \mathbb{P}_{\mathbf{Z}}$, consider the following set:

$$\mathcal{C}_h = \{j : \mathbf{Z}_j \text{ was generated from } \mathbb{P}_h \text{ for } 1 \leq j \leq N\} \tag{6}$$

with $h \in \{1, 2\}$. Clearly, the set $\mathcal{C}_h$ depends on the sample size $N$. The components of the mixture distribution $\mathbb{P}_{\mathbf{Z}}$ and the sets $\mathcal{C}_h$ for $h = 1, 2$ are *unknown*, and the problem we are addressing here is the estimation of these sets. However, we assume $\mathbb{P}_{\mathbf{Z}}$ and the sets $\mathcal{C}_1$ and $\mathcal{C}_2$ to be *known* in this section to build the fundamental idea behind using the proposed transformation for clustering of GPs.

Let $V_d$ with $d \in \mathbb{N}$ denote the sequence of $d$-dimensional subspaces generated by the $d$ eigenfunctions associated with the $d$ largest eigenvalues of $\Sigma^{\mathbf{Z}}$ (recall the discussion in Remark 2.1.4). Consider a sub-sample of size two from $\mathbf{Z}_1, \ldots, \mathbf{Z}_N$ generated from $\mathbb{P}$. To simplify our notation, we denote them to be $\mathbf{Z}_1$ and $\mathbf{Z}_2$, respectively. They are independent, with $\mathbb{P}_{\mathbf{Z}_1} = \mathbb{P}_h$ and $\mathbb{P}_{\mathbf{Z}_2} = \mathbb{P}_k$ for $h, k \in \{1, 2\}$. The clustering procedure that we propose is based on the behavior of the transformation $D_d^{\Sigma^{\mathbf{Z}}}(\mathbf{Z}_1, \mathbf{Z}_2)$, which is stated below in Theorem 2.2.

**Theorem 2.2** *(a) Assume that $h = k \in \{1, 2\}$. Define $S_d^h := (\Sigma_d^{\mathbf{Z}})^{-1/2}(2\Sigma_d^{\mathbf{Z}_h})(\Sigma_d^{\mathbf{Z}})^{-1/2}$ for $d \in \mathbb{N}$, and assume that $L_S^h = \lim_d \frac{1}{d} trace(S_d^h)$ exists. Then, $L_S^h$ is finite and*

$$D_d^{\Sigma_d^{\mathbf{Z}}}(\mathbf{Z}_1, \mathbf{Z}_2) \xrightarrow{P} L_S^h \text{ as } d \to \infty. \tag{7}$$

*(b) Assume that $h \neq k \in \{1, 2\}$. Then, $L_\mu^{hk} = \lim_d D_d^{\Sigma_d^{\mathbf{Z}}}(\mu^{\mathbf{Z}_h}, \mu^{\mathbf{Z}_k}) = 0$.*

*Define $S_d^{hk} := (\Sigma_d^{\mathbf{Z}})^{-1/2}(\Sigma_d^{\mathbf{Z}_h} + \Sigma_d^{\mathbf{Z}_k})(\Sigma_d^{\mathbf{Z}})^{-1/2}$ for $d \in \mathbb{N}$, and assume that $L_S^{hk} = \lim_d \frac{1}{d} trace(S_d^{hk})$ exists. Then, $L_S^{hk}$ is finite and*

$$D_d^{\Sigma_d^{\mathbf{Z}}}(\mathbf{Z}_1, \mathbf{Z}_2) \xrightarrow{P} L_S^{hk} \text{ as } d \to \infty. \tag{8}$$

**Remark 2.2.1** The structure of the covariance $\Sigma^{\mathbf{Z}}$ stated in equation (5) imposes some restrictions on the associated constants as stated in part (b) of Theorem 2.2. In particular, the fact that $L_S^h$ and $L_S^{hk}$ are finite implies that assumption (2) in Theorem 2.1 always holds for the sequence of matrices $\{S_d^h\}_{d \in \mathbb{N}}$ and $\{S_d^{hk}\}_{d \in \mathbb{N}}$ with $h, k \in \{1, 2\}$.

**Remark 2.2.2** It follows from part (b) of Theorem 2.2 that the statistic we propose is useless for cluster analysis in the homoscedastic case (independently of the difference between $\mu^{\mathbf{Z}_1}$ and $\mu^{\mathbf{Z}_2}$) because if $\Sigma^{\mathbf{Z}_1} = \Sigma^{\mathbf{Z}_2}$, then $L^{12} = L_S^1 = L_S^2$. A possibility is to modify the statistic $D_d^{\Sigma_d^{\mathbf{Z}}}(\mathbf{z}_1, \mathbf{z}_2)$ so that the value of the transformation $D_d^{\Sigma_d^{\mathbf{Z}}}(\mu^{\mathbf{Z}_1}, \mu^{\mathbf{Z}_2})$ increases with $d$. Our proposal is to use

$$D_d^{\Sigma_d^{\mathbf{Z}}, r}(\mathbf{u}, \mathbf{v}) := \frac{1}{d} \left\| ((\Sigma_d^{\mathbf{Z}})^{-1/2})^r (\mathbf{u} - \mathbf{v})_d \right\|^2 = \frac{1}{d} \sum_{i=1}^d \frac{(u_i - v_i)^2}{\lambda_i^r} \text{ with } r \in \mathbb{N}.$$

Discussion of this transformation, and some numerical results are included in Section L of Appendix II.

To simplify notation and avoid technicalities with empty classes, we additionally assume that the observations whose indices belong to the sets $\mathcal{C}_1 = \{1, \ldots, N_1\}$ and $\mathcal{C}_2 = \{N_1 + 1, \ldots, N\}$ with $N = N_1 + N_2$ and $N_1, N_2 > 0$, were generated by $\mathbb{P}_1$ and $\mathbb{P}_2$, respectively. In practice, these sets are *unknown* and in fact our aim is their estimation. We begin with this simplifying assumption for ease of notation, and to obtain a clearer exposition of the proposed methodology.

Define the $N \times N$ matrix $\Gamma_d$, whose $(i, j)$-th element is

$$\Gamma_d(\mathbf{Z}_i, \mathbf{Z}_j) = \gamma_{ij}^d = \frac{1}{N-2} \sum_{t=1, \ t \neq i, j}^N \left[ D_d^{\Sigma_d^{\mathbf{Z}}}(\mathbf{Z}_t, \mathbf{Z}_i) - D_d^{\Sigma_d^{\mathbf{Z}}}(\mathbf{Z}_t, \mathbf{Z}_j) \right]^2 \tag{9}$$

for $1 \leq i, j \leq N$. Theorem 2.2 and the fact that $t \neq i, j$ in (9) give us the following:

$$\gamma_{ij}^d \xrightarrow{P} \begin{cases} 0, & \text{if } i, j \in \mathcal{C}_h \text{ for } h = 1, 2, \\ \gamma_{hk}, & \text{if } i \in \mathcal{C}_h \text{ and } j \in \mathcal{C}_k, \text{with } h \neq k \in \{1, 2\}, \end{cases} \tag{10}$$

as $d \to \infty$. Here,

$$\gamma_{hk} = \frac{N_h - 1}{N - 2}(L_S^h - L_S^{hk})^2 + \frac{N_k - 1}{N - 2}(L_S^k - L_S^{kh})^2.$$

Note that $N_1$ and $N_2$ are fixed here. Combining the fact stated above in (10), as $d \to \infty$, we obtain

$$\Gamma_d \xrightarrow{P} \Gamma := \begin{bmatrix} \mathbf{0}_{N_1}\mathbf{0}_{N_1}^T & \gamma_{12}\mathbf{1}_{N_1}\mathbf{1}_{N_2}^T \\ \gamma_{21}\mathbf{1}_{N_2}\mathbf{1}_{N_1}^T & \mathbf{0}_{N_2}\mathbf{0}_{N_2}^T \end{bmatrix}. \tag{11}$$

Let $\beta_i^d$ and $\beta_i$ (for $1 \le i \le N$) denote the eigenvalues corresponding to the matrices $\Gamma_d$ and $\Gamma$, respectively. Define the following quantities

$$K_d = \sum_{i=1}^N I(|\beta_i^d| > a_d) \text{ and } K_0 = \sum_{i=1}^N I(|\beta_i| > 0), \tag{12}$$

with $\{a_d\}_{d \in \mathbb{N}}$ decreasing to 0 as $d \to \infty$ (at an appropriate rate), and $I$ is the indicator function. The constant $K_0$ clearly equals 2 for the limiting $N \times N$ matrix $\Gamma$ stated in (11), and hence, correctly identifies the true number of clusters.

**Proposition 2.3** *Assume $N_1, N_2 \ge 1$ are fixed. Under the assumptions of Theorem 2.2, if $L_S^{12} \ne L_S^1$ and $L_S^{21} \ne L_S^2$, then there exists a sequence $\{a_d\}_{d \in \mathbb{N}} \subset \mathbb{R}^+$ such that $a_d \to 0$ and $K_d \xrightarrow{P} 2$ as $d \to \infty$.*

This now implies that we can correctly identify the true number of clusters asymptotically as $d \to \infty$. The structure of the matrix $\Gamma$ in (11) is straight forward because of the simplifying assumption on the sets $\mathcal{C}_1$ and $\mathcal{C}_2$. However, this is not a requirement and we will drop it. Proposition 2.3 holds more generally for *any permutation* of the data points $\mathbf{Z}_1, \ldots, \mathbf{Z}_N$. In fact, if the sets $\mathcal{C}_1$ and $\mathcal{C}_2$ are *unknown*, then the rows/columns of the $\Gamma$ matrix will be permuted accordingly. But, the underlying structure remains the same and Proposition 2.3 continues to hold. As a followup of our previous result, we now prove that if any standard clustering method is used on the $\Gamma_d$ matrix, then we can *perfectly cluster* all the observations asymptotically (as $d \to \infty$) because of the structure of the $\Gamma$ matrix stated in (11).

**Definition 2.4** *A clustering method can be defined as a map from $\mathbb{H}$ to the set $\{1, \ldots, J\}$. Consider the sequence of maps $\{\psi_d\}_{d \in \mathbb{N}}$ and a second map $\phi$. A measure of distance between two clusterings based on the Rand index (see p. 847 of Rand (1971)) is defined as follows:*

$$\mathbb{R}_{d,N} = \frac{1}{\binom{N}{2}} \sum_{1 \le i < j \le N} I\big[I[\psi_d(\mathbf{z}_i) = \psi_d(\mathbf{z}_j)] + I[\phi(\mathbf{z}_i) = \phi(\mathbf{z}_j)] = 1\big],$$

*for a fixed $N \ge 2$.*

Let $\phi$ be the map which gives the true labels, i.e., $\phi(\mathbf{x}_j) = h$ for $j \in \mathcal{C}_h$ with $h \in \{1, 2\}$. We can construct a data based $\psi_d$ by directly applying any clustering technique on the rows or columns of the matrix $\Gamma_d$. Here, we use the $k$-means algorithm on the rows of $\Gamma_d$.

Mathematically, the $k$-means algorithm finds $J$ groups (say, $\mathcal{G}_1, \ldots, \mathcal{G}_J$) with centers $\mathbf{c}_1, \ldots, \mathbf{c}_J$ such that $\phi(\mathcal{G}_1, \ldots, \mathcal{G}_J) = \sum_{h=1}^J \sum_{\{i:\mathbf{x}_i \in \mathcal{G}_h\}} \|\mathbf{x}_i - \mathbf{c}_h\|^2$ is minimized. The asymptotic properties of the matrix $\Gamma_d$ as $d \to \infty$ (stated above in (11)) imply that differences in the limiting constants should yield *perfect clustering*. Our next result proves label consistency for this $k$-means algorithm when $J = 2$.

**Theorem 2.5** *Assume $J = 2$ and $\gamma_{12} > 0$. Further, assume that the conditions in Theorem 2.2 and Proposition 2.3 hold. Then, the clusters will be perfectly identifiable, i.e., $\mathbb{R}_{d,N} \xrightarrow{P} 0$ as $d \to \infty$.*

**Remark 2.5.1** The well-known Rand index (*a measure of similarity*) is usually defined as $1 - \mathbb{R}_{d,N}$. As a consequence, Theorem 2.5 implies that the usual Rand index goes to one as $d \to \infty$.

**Remark 2.5.2** The structure of the $N \times N$ symmetric matrix $\Gamma$ stated in (11) continues to hold, and will lead us to *perfect clustering* for every value of $J \geq 2$ if enough distinct $\gamma_{ij}$'s exist. In particular, this happens if we have some $h$ with $1 \leq h \leq J$ such that $\gamma_{hk} \neq \gamma_{hk'}$ for every $1 \leq k \neq k' \leq J$, but other possibilities exist as well.

Moreover, the procedure described in Proposition 2.3 also works fine with the limit equal to $K_0$ (which is the rank of $\Gamma$). One may be tempted to think that it generally coincides with $J$. But, this is true only for $J \leq 3$ and may be different for $J \geq 4$ (as shown in Lemma F.1 of Appendix F). The proof of Lemma F.1 further shows that the condition under which $\text{rank}(\Gamma) < J$ is quite restrictive. Thus, for simplicity, our proposal is to estimate the number of clusters $J$ using $K_d$ (defined in equation (12) above).

### 2.1.1 Example with GPs

If we assume that $\Sigma^{\mathbf{Z}_2} = a\Sigma^{\mathbf{Z}_1}$ (for some $a > 0$), then we have the following expressions for the scale constants stated in Theorem 2.2:

$$L_S^1 = \frac{2}{\pi_1 + \pi_2 a}, \ L_S^2 = \frac{2a}{\pi_1 + \pi_2 a} \text{ and } L_S^{12} = \frac{1+a}{\pi_1 + a\pi_2}.$$

Thus, it is possible to identify perfectly the clusters as long as $a \neq 1$, since this implies that $\gamma_{12}$ and $\gamma_{21}$ both are positive quantities.

### 2.1.2 Uniform Convergence

In Theorem 2.2, we have proved consistency for finite sets of data points corresponding to the transformation $D_d^{\Sigma_d^{\mathbf{Z}}}(\mathbf{Z}_1, \mathbf{Z}_2)$ defined in (1). We now prove the uniform (on the random sample) convergence of this function as $N \to \infty$. This result will be useful in establishing a second result on uniform convergence, which we state in the next section.

**Theorem 2.6** *Assume the conditions in Theorem 2.2, and let $\{d_N\} \subset \mathbb{N}$ be such that $d_N \to \infty$ as $N \to \infty$. Then, we have the following.*

a) *For $h \in \{1, 2\}$, let $\alpha_{d_N} = (\alpha_1^{d_N}, \ldots, \alpha_{d_N}^{d_N})^T$ be the eigenvalues of $S_{d_N}^h$ with $d_N \in \mathbb{N}$. If*

$$\log N = o\left(\frac{d_N}{\|\alpha_{d_N}\|_\infty}\right), \tag{13}$$

*then it happens that*

$$\sup_{\mathbf{Z}_1, \mathbf{Z}_2 \in \mathcal{C}_h^N} \left| D_{d_N}^{\Sigma_{d_N}^{\mathbf{Z}}}(\mathbf{Z}_1, \mathbf{Z}_2) - L_S^h \right| \xrightarrow{P} 0 \ as \ N \to \infty. \tag{14}$$

b) *For any $h \neq k \in \{1, 2\}$, let $\alpha_{d_N} = (\alpha_1^{d_N}, \ldots, \alpha_{d_N}^{d_N})^T$ be the eigenvalues of $S_{d_N}^{hk}$ with $d_N \in \mathbb{N}$. If*

$$\log N = o\left(\frac{d_N}{\|\alpha_{d_N}\|_\infty}\right), \tag{15}$$

*then it happens that*

$$\sup_{\mathbf{Z}_1 \in \mathcal{C}_h^N, \mathbf{Z}_2 \in \mathcal{C}_k^N} \left| D_{d_N}^{\Sigma_{d_N}^{\mathbf{Z}}}(\mathbf{Z}_1, \mathbf{Z}_2) - L^{hk} \right| \xrightarrow{P} 0 \ as \ N \to \infty. \tag{16}$$

**Remark 2.6.1** Assumption (2) holds here, so $\frac{\|\alpha_{d_N}\|_\infty}{d_N} = \frac{1}{d_N} \max_{1 \leq i \leq d_N} \alpha_i^{d_N} \to 0$ as $N \to \infty$. Thus, if we take $d_N$ growing fast enough, then it is assured that assumptions (13) and (15) hold. The structure of the matrices $S_d^h$ and $S_d^{hk}$ for $d \in \mathbb{N}$ with $h \neq k \in \{1, 2\}$ implies that a sufficient condition is $\log N = o(d_N)$ (also see Proposition H.1 in Appendix H).

## 3 Transformations with Estimated Gaussian Distributions

In this section, we will discuss the steps to implement the procedure described in Section 2. In practice, the involved distributions and all the associated quantities *need to be estimated from the data*. Here, $\mathbf{Z}$ will denote a random element with distribution the GP mixture $\pi_1 \mathbb{P}_1 + \pi_2 \mathbb{P}_2$.

Let $\phi_j^{\mathbf{Z}}(t)$ with $t \in [0,1]$ and $\lambda_j^{\mathbf{Z}}$ for $j \in \mathbb{N}$ denote the eigenfunctions and eigenvalues of $\Sigma^{\mathbf{Z}}$, respectively. We will now make the following assumptions:

A.1 $\sup_{t \in [0,1]} E[\mathbf{Z}^4(t)] < \infty$.

A.2 Assume that $\lambda_1^{\mathbf{Z}} > \lambda_2^{\mathbf{Z}} > \cdots > 0$ satisfying $\sum_{j=1}^{\infty} \lambda_j^{\mathbf{Z}} < \infty$.

It is well-known that assumption *A.2* implies $\{\phi_j^{\mathbf{Z}}\}_{j \in \mathbb{N}}$ forms an orthonormal basis of $\mathbb{H}$.

To estimate $\Sigma^{\mathbf{Z}}$ and its eigenvalues and eigenfunctions, we will use the corresponding empirical quantities. Suppose that we have a simple random sample $\mathbf{Z}_1, \ldots, \mathbf{Z}_N$ taken from $\mathbb{P}_{\mathbf{Z}}$. Given $s, t \in [0,1]$, we define

$$\hat{\Sigma}^{\mathbf{Z}}(s,t) = \frac{1}{N} \sum_{i=1}^{N} [\mathbf{Z}_i(s) - \overline{\mathbf{Z}}_N(s)][\mathbf{Z}_i(t) - \overline{\mathbf{Z}}_N(t)],$$

where $\bar{\mathbf{Z}}_N(t) = \frac{1}{N} \sum_{i=1}^{N} \mathbf{Z}_i(t)$. Consider the corresponding families $\hat{\lambda}_1^{\mathbf{Z}} \geq \hat{\lambda}_2^{\mathbf{Z}} \geq \cdots$ and $\hat{\phi}_1^{\mathbf{Z}}, \hat{\phi}_2^{\mathbf{Z}}, \ldots$ of its eigenvalues and eigenvectors, respectively. Note that $\hat{\Sigma}^{\mathbf{Z}}$ as well as all the $\hat{\lambda}_j^{\mathbf{Z}}$'s and $\hat{\phi}_j^{\mathbf{Z}}$'s depend on $N$.

With a finite sample, we cannot estimate the entire (infinite) collection of eigenvalues and eigenvectors. Thus, we follow the work of Delaigle and Hall (2012) and Hall and Hosseini-Nasab (2006), and select a non-random decreasing sequence $\eta_N$ going to zero slowly enough as to satisfy $\lim_N N^{1/5} \eta_N = \infty$. We define

$$\hat{R}_N^{\mathbf{Z}} = \inf\{j : \hat{\lambda}_j^{\mathbf{Z}} - \hat{\lambda}_{j+1}^{\mathbf{Z}} < \eta_N\} - 1. \tag{17}$$

This definition implies that $\hat{\lambda}_j^{\mathbf{Z}} \geq \eta_N$ for every $j \leq \hat{R}_N^{\mathbf{Z}}$. Moreover, we also need that the theoretical eigenvalues are reasonably well separated. To obtain this, given $\delta > 0$, we further define

$$R_N^{\mathbf{Z}} = \inf\{j : \lambda_j^{\mathbf{Z}} - \lambda_{j+1}^{\mathbf{Z}} < (1 + \delta)\eta_N\} - 1. \tag{18}$$

We now state empirical analogues of the results stated in Section 2.1.

### 3.1 Consistency of Clustering

Let $\mathbf{Z}_1, \ldots, \mathbf{Z}_N$ be a simple random sample taken from the GP mixture distribution $\mathbb{P}_{\mathbf{Z}}$. Now, $\mathbb{P}_{\mathbf{Z}}$ and the sets $\mathcal{C}_1$ and $\mathcal{C}_2$ (containing information on the class labels defined in (6)) are *unknown*. Extensions of Theorems 2.2 and 2.6 to Theorems 3.1 and 3.2 are presented below. The following results will be based on the analysis of the map $\hat{D}_{\hat{R}_N}(\mathbf{u}, \mathbf{v})$, which is the transformation $D_d^{\Sigma_d^{\mathbf{Z}}}(\mathbf{u}, \mathbf{v})$ defined in (1) with $d = \hat{R}_N$ (defined in (17)), and the pooled covariance matrix $\Sigma_{\hat{R}_N}^{\mathbf{Z}}$ which is estimated by $\hat{\Sigma}_{\hat{R}_N}^{\mathbf{Z}}$ (sample covariance of the full sample). Here, $\hat{D}_{\hat{R}_N}$ is an abridged notation of $D_{\hat{R}_N}^{\hat{\Sigma}_{\hat{R}_N}^{\mathbf{Z}}}$. The first result is related to the consistency of the transformation on finite sets.

**Theorem 3.1** *Let assumptions A.1 and A.2 and those in Theorem 2.2 hold.*

*(a) If $h = k \in \{1, 2\}$, then*

$$\hat{D}_{\hat{R}_N}(\mathbf{Z}_1, \mathbf{Z}_2) \xrightarrow{P} L_S^h \text{ as } N \to \infty. \tag{19}$$

*(b) If $h \neq k \in \{1, 2\}$, then*

$$\hat{D}_{\hat{R}_N}(\mathbf{Z}_1, \mathbf{Z}_2) \xrightarrow{P} L_S^{hk} \text{ as } N \to \infty. \tag{20}$$

We need an increasing sample size in order to estimate the parameters consistently. Thus, it is desirable to be able to cluster the increasing number of data points, asymptotically without error. The only way to achieve this is to get some kind of uniform convergence in (19) and (20) when the sample size increases. This is the purpose of Theorem 3.2, which gives us clear evidence that using this transformation would lead to asymptotic perfect separation in the empirical case as well.

**Theorem 3.2** *Let us assume all the conditions in Theorem 2.6 with* $\log N = o(R_N^{\mathbf{Z}})$ *in (18).*

*(a) For* $h \in \{1, 2\}$*, we have that*

$$\sup_{\mathbf{Z}_1, \mathbf{Z}_2 \in \mathcal{C}_h^N} \left| \hat{D}_{\hat{R}_N}(\mathbf{Z}_1, \mathbf{Z}_2) - L_S^h \right| \xrightarrow{P} 0 \ as \ N \to \infty.$$

*(b) For any* $h, k \in \{1, 2\}$ *with* $h \neq k$*, we have that*

$$\sup_{\mathbf{Z}_1 \in \mathcal{C}_h^N, \mathbf{Z}_2 \in \mathcal{C}_k^N} \left| \hat{D}_{\hat{R}_N}(\mathbf{Z}_1, \mathbf{Z}_2) - L_S^{hk} \right| \xrightarrow{P} 0 \ as \ N \to \infty.$$

**Remark 3.2.1** Clearly, Theorem 3.1 follows from Theorem 3.2. But, the conditions required for proving the former are weaker and hence, we state it as a separate result.

**Remark 3.2.2** (*Asymptotic perfect identification of clusters*) Recall the matrix $\Gamma_d$ from (9) with $d \in \mathbb{N}$. Now, consider the matrix $\hat{\Gamma}_{\hat{R}_N}$, which is obtained by replacing $\gamma_{ij}^d$s in the matrix $\Gamma_{\hat{R}_N}$ with their estimated values $\hat{\gamma}_{ij}^{\hat{R}_N}$, i.e., $\hat{\gamma}_{ij}^{\hat{R}_N} = \hat{D}_{\hat{R}_N}(\mathbf{Z}_i, \mathbf{Z}_j)$ with $1 \leq i \neq j \leq N$. Define $v_{12} = \pi_1 \left| L_S^1 - L_S^{12} \right|^2 + \pi_2 \left| L_S^2 - L_S^{21} \right|^2$. Fix $\epsilon > 0$. Theorem 3.2 implies that with probability converging to one as $N \to \infty$, we have

- if $\mathbf{Z}_i, \mathbf{Z}_j \in \mathcal{C}_h$ for $h \in \{1, 2\}$, then $\left| \hat{\gamma}_{ij}^{\hat{R}_N} \right| \leq 4\epsilon^2$,

- if $\mathbf{Z}_i \in \mathcal{C}_h, \mathbf{Z}_j \in \mathcal{C}_k$ for $h \neq k \in \{1, 2\}$, then $\left| \hat{\gamma}_{ij}^{\hat{R}_N} - v_{12} \right| \leq H\epsilon$,

for some $H > 0$. Consequently, if $v_{12} > 0$, then the elements in $\hat{\Gamma}_d$ will be clustered into two well-separated clusters: one around 0 and another one around $v_{12}$ with probability converging to one.

Similarly, let $\mathbb{P}_{\mathbf{Z}}$ be a mixture of $J(> 2)$ components and denote

$$v_{hk} := \pi_h \left| L_S^h - L_S^{hk} \right|^2 + \pi_k \left| L_S^k - L_S^{kh} \right|^2$$

with $1 \leq h \neq k \leq J$. For positive and distinct $v_{hk}$s, the elements in the matrix $\hat{\Gamma}_d$ will be perfectly clustered into $1 + \binom{J}{2}$ well separated clusters: one of them around the point 0 and the remaining around the values $v_{hk}$ (for $h < k$) with probability converging to one as $N \to \infty$. Therefore, asymptotically, the sequence of matrices $\{\hat{\Gamma}_{\hat{R}_N}\}_{N \in \mathbb{N}}$ will contain enough information to perfectly cluster all the data points in a sample.

## 3.2 Implementation Issues

We are given a sample of data points without the labels. Here, we consider the $N \times N$ estimated matrix $\hat{\Gamma}_N$ with the $(i, j)$-th element as $\hat{D}_{\hat{R}_N}(\mathbf{Z}_i, \mathbf{Z}_j)$ (which is just the empirical version of $D_d^{\Sigma_d^{\mathbf{Z}}}(\mathbf{Z}_i, \mathbf{Z}_j)$ based on the pooled sample covariance) for $1 \leq i, j \leq N$ and apply any clustering procedure on its rows (or, columns). Note again that we *do not need to estimate* the unknown constants $L_S^h$ and $L_S^{hk}$ for $h, k \in \{1, 2\}$ (stated in Theorem 3.1) for the implementation of our clustering procedure. The expressions related with $K_d$ and $\hat{R}_N$ are not used in our implementation (see Table 3.2 below for the main differences between theory and practice). In Section 4.2, we give complete details of the implementation of our procedure.

Table 3.1: Some key differences in our theoretical assumptions and implementations.

| Quantity | Theoretical Assumption | Implementation |
|---|---|---|
| $\hat{R}_N$ | involves $\hat{\lambda}_j$s and goes to zero as $N \to \infty$ | estimated using cross-validation in clustering (see Wang (2010) for more details) |
| $K_d$ | involves $\beta_i$s and goes to zero as $d \to \infty$ | involves $\hat{\beta}_i$s and estimated using the `optishrink` function from the R package `denoiseR` |

## 4 Analysis of Simulated Datasets

For our simulation study, we consider two class problems ($J = 2$). We generated data on a discrete grid of 100 equi-spaced points in the unit interval $[0, 1]$ from four different simulation models, which are described below. Fix $s > 0$.

I. Define $X_h(t) = \sum_{j=1}^{40}(\lambda_{hj}^{1/2}Z_{hj} + \mu_{hj})\phi_j(t)$ with $t \in [0, 1]$ and $h = 1, 2$. Here, the $Z_{hj}$s are independent standard normal (i.e., $N(0, 1)$) random variables, $\phi_j(t) = \sqrt{2}\sin(\pi jt)$ with $t \in [0, 1]$ and $1 \le j \le 40$. Also, $\mu_{hj} = 0$ for $j > 6$, and we set the other components equal to $(0, -0.5, 1, -0.5, 1, -0.5)^T$ and $(0, -0.75, 0.75, -0.15, 1.4, 0.1)^T$ for $k = 1, 2$, and $\lambda_{1j} = 1/j^2$ and $\lambda_{2j} = s/j^2$ for $1 \le j \le 40$. This model is from the paper `Delaigle and Hall (2012)`.

II. In this example, $X_1 \sim B$ and $X_2 \sim \mu + sB$ with $\mu(t) = Gt$ for $t \in [0, 1]$ and $G \sim N(0, 4)$ independent of $B$. Here, $B$ is the standard Brownian bridge, i.e., a centered GP with $\sigma_{ij} = \min(t_i, t_j) - t_i t_j$ for $t_i, t_j \in [0, 1]$ and $i, j \in \mathbb{N}$.

   Since $E[X_2(t)] = E[Gt] = 0$ for $t \in [0, 1]$, the *differences in mean never appear* in this setting. In fact, the inclusion of $\mu$ modifies the covariances because if $0 < t_i < t_j < 1$, then the independence between $G$ and $B$ yields the following:

   $$E[X_2(t_i)X_2(t_j)] = 4t_i t_j + s^2 t_i(1 - t_j).$$

   This model is from the paper `Berrendero et al (2018)`.

III. Let $X_h = \mu_h + \sum_{j=1}^{50} \xi_{hj}\lambda_{hj}^{1/2}\phi_j$ for $h = 1, 2$. Here, $\xi_{hj}$s are i.i.d. $N(0, 1)$, $\mu_1 = 0$ and $\mu_2(t) = t$ with $t \in [0, 1]$, $\lambda_{1j} = e^{-j/3}$ and $\lambda_{2j} = \sqrt{s}e^{-j/3}$ for $1 \le j \le 50$, and $\phi_{2i-1} = \sqrt{2}\sin(2i\pi t)$ and $\phi_{2i} = \sqrt{2}\cos(2i\pi t)$ for $1 \le i \le 25$ with $t \in [0, 1]$. This model is from the paper `Dai et al (2017)`.

IV. This problem consists of two Brownian motions defined in the closed interval $[0, 1]$ with mean functions $\mu^{Z_1}(t) = 20t^{1.1}(1 - t)$ and $\mu^{Z_2}(t) = 20t(1 - t)^{1.1}$, respectively, for $t \in [0, 1]$. For the first class, the eigenfunctions are $\phi_j(t) = \sqrt{2}\sin((j - 0.5)\pi t)$ and associated eigenvalues are $\lambda_{1j} = 1/(\pi(j - 0.5))^2$ for $1 \le j \le 15$. The second class is similar to the first one, but the eigenvalues are multiplied by $\sqrt{s}$ (i.e., $\lambda_{2j} = \sqrt{s}\lambda_{1j} = \sqrt{s}/(\pi(j - 0.5))^2$) for $1 \le j \le 15$. This model is from the paper `Galeano et al (2015)`.

We set $s = 1$ for `location only` problems. In `location and scale` problems, we fixed $s = 3$, while for `scale only` problems the mean functions $\mu^{Z_1}$ and $\mu^{Z_2}$ were set to be the constant function 0 and $s = 3$ was retained.

### 4.1 Choice of $d$

A critical issue here is selection of the optimal dimension of the projected space for a given a set of data points (i.e., a fixed value of $(N_1, N_2)$ or $N$). Let us recall Theorem 3.1. According to this result, we expect the rows of the matrix $\hat{\Gamma}_d$ to form two clearly separated clusters depending on the class label of the observation for

large values of $d$. To demonstrate this, we construct a sequence of images which shows how this separation varies with increasing values of $d$.

We generated samples of size 250 from each of the two classes for the 'scale case' of Example II. For the purpose of demonstration, the first 250 observations correspond to the first GD, while the next 250 observations to the second. Figure 1 below shows the heatmap for increasing values of $d$, and we observe the best concentration at $d = 80$. However, some noise in the off-diagonal submatrices for $d = 80$ (compared to $d = 60$) makes us to consider that the optimum could be somewhere between the values 60 and 80.

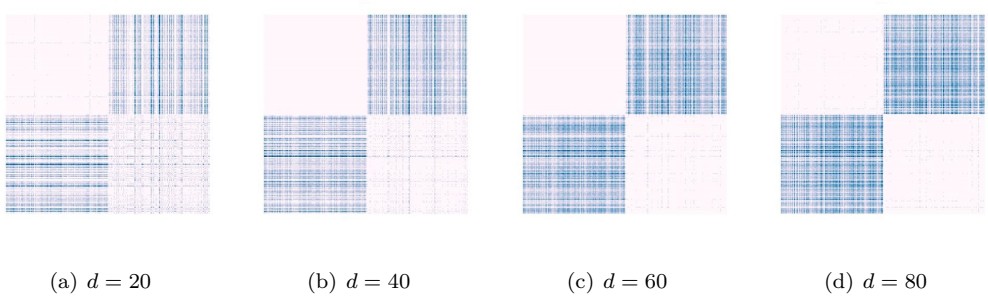

(a) $d = 20$        (b) $d = 40$        (c) $d = 60$        (d) $d = 80$

Figure 1: Heatmap of $\hat{\Gamma}_d$ for varying values of $d$.

Clearly, the choice of $d$ is quite important as $d$ is the dimension of the subspace where we project our observations (for a fixed sample size). We observe from Figure 1 that its estimation is quite crucial. The next subsection contains further details on the choice of $d$.

## 4.2 Clustering Procedure in Practice

To implement the clustering method, we use cross-validation (CV) to choose the dimension $d$ suitably. We use the idea developed by Wang (2010), which we state briefly here. Given $B \in \mathbb{N}$, split the data into three random subsets (say, $S_{1b}$, $S_{2b}$ and $S_{3b}$) each of equal size for $1 \leq b \leq B$. For each value of $b$, treat the data points in $S_{1b}$ and $S_{2b}$ as the training sets, and $S_{3b}$ as the validation set. For a fixed value of $d$ and given a clustering algorithm, the two training sets $S_{1b}$ and $S_{2b}$ are used to construct two cluster assignments. An appropriate distance between these two cluster assigments (say, $\mathbb{D}$) is computed based on the validation set $S_{3b}$ (see Section 2 of Wang (2010) for more details). We repeat this partitioning $B(= 50)$ times and average it over these $B$ samples to get $\hat{\mathbb{D}}_d^{CV}$. Define $\hat{d}_{CV} = \arg\min_{2 \leq d \leq N} \hat{\mathbb{D}}_d^{CV}$. A pseudo-code for this procedure is given in Algorithm 2 (see Section O of the Appendix).

Recall the structure of the $\Gamma$ matrix stated in (11), and also see Figure 1. As mentioned in Section 3.2, the number of clusters were estimated using the quantity $K_d$ described in Section 2.1 (see (12)). To implement the procedure in practice, one needs to estimate the sequence $\{a_d\}_{d \in \mathbb{N}}$. We have used the function `optishrink` available in the `R` package `denoiseR`. This function extracts a *low-rank signal from Gaussian noisy data using the optimal shrinker of singular values.* The low rank structure of the $\Gamma$ matrix motivates us to directly apply this function on $\hat{\Gamma}_d$. A pseudo-code (alongwith stepwise computational complexity) of our methods is given in Algorithm 1 (see Section O of the Appendix).

Our overall implementation yielded quite desirable results in our numerical study (see Tables 4.1 and 4.2 below). We can apply any clustering method on the transformed data $\hat{\Gamma}_d$. In addition to the $k$-means algorithm (CD-$k$-means) discussed in Theorem 2.5, we considered spectral clustering (CD-spectral) and Gaussian mixture models (CD-mclust). Here, the acronym CD corresponds to our proposed transformation. One may refer to the book by Hastie et al (2009) for details on these three popular clustering methods. The `R` codes for our methods are available here: GP clustering.

We considered several methods for comparison. The first three methods are those that we have used on the matrix $\hat{\Gamma}_d$ but applied directly to functional data, namely, the $k$-means algorithm ($k$-means), spectral clustering (spectral) and Gaussian mixture models (mclust). Several competent methods for functional

clustering using functional mixed mixture models are implemented in the function `funcit` from the `R` package `funcy`. We report this method as funclust. The methodology developed by Chiou and Li (2007) is available in the function `FClust` from the `R` package `fdaspace` using two clustering techniques 'EMcluster' (CL1) and 'kCFC' (CL2). We have reported the minimum value, and stated it as CL. In Delaigle et al (2019), the authors developed functional clustering based on the $k$-means using basis functions. We implemented this method for two choices of the basis functions, namely, Haar and PC, and reported the best result among these two (we call it DHP). We have not used the DB2 basis for our comparisons because it requires the grid points to be of a power of 2. The DHP method is available from the journal website, and we used those `Matlab` codes for our comparisons.

We conducted simulations based on models I to IV, which were introduced in the beginning of Section 4. We did not consider the `location only` scenario as our proposed method is useless in such cases (recall part (c) of Theorem 2.2). However, we have some discussion and additional results in Section L in Appendix II for this scenario. The sample size of each class was set to be 250. Our experiment was replicated 100 times, and the results are reported in Tables 4.1 and 4.2 below. To measure the similarity between two cluster assignments, we computed the adjusted Rand index using the function `RRand` in the `R` package `phyclust`. One minus the adjusted Rand index (we call it *adjusted Rand distance*) is reported in tables below, where the minimum is marked in **bold** and the second lowest is in *italics*.

It is worth noting that all the competing methods require the number of clusters as an input variable, and we have run these methods with $k = 2$ (the true number of clusters). However, when applying the CD procedure we have estimated the number of clusters following the procedure described above. We obtained the correct value in more than 99% of the cases (across all four examples for both scenarios) in our simulation study.

Table 4.1: Adjusted Rand distances for different GPs with differences in locations and scales (with standard error in brackets).

| Ex. | $k$-means | spectral | mclust | funclust | CL | DHP | CD | | |
|---|---|---|---|---|---|---|---|---|---|
| | | | | | | | $k$-means | spectral | mclust |
| I | 0.0632 | 0.0280 | 1.0000 | 0.1541 | 0.0239 | 0.0818 | 0.0654 | *0.0028* | **0.0001** |
| | (0.0007) | (0.0016) | (0.0000) | (0.0017) | (0.0007) | (0.0025) | (0.0010) | (0.0002) | (0.0001) |
| II | 0.5377 | 0.4210 | 0.9700 | 0.8222 | 0.5767 | 0.5149 | *0.2104* | 0.3464 | **0.0738** |
| | (0.0046) | (0.0042) | (0.0017) | (0.0027) | (0.0045) | (0.0049) | (0.0030) | (0.0045) | (0.0014) |
| III | 0.4250 | 0.4460 | 1.0000 | 0.3858 | 0.2891 | 0.4137 | *0.1367* | 0.2596 | **0.0250** |
| | (0.0017) | (0.0048) | (0.0000) | (0.0003) | (0.0000) | (0.0054) | (0.0024) | (0.0042) | (0.0016) |
| IV | 0.4945 | 0.3788 | 1.0000 | 0.3975 | 0.1833 | 0.1379 | *0.0316* | **0.0000** | **0.0000** |
| | (0.0056) | (0.0047) | (0.0000) | (0.0011) | (0.0000) | (0.0033) | (0.0001) | (0.0000) | (0.0000) |

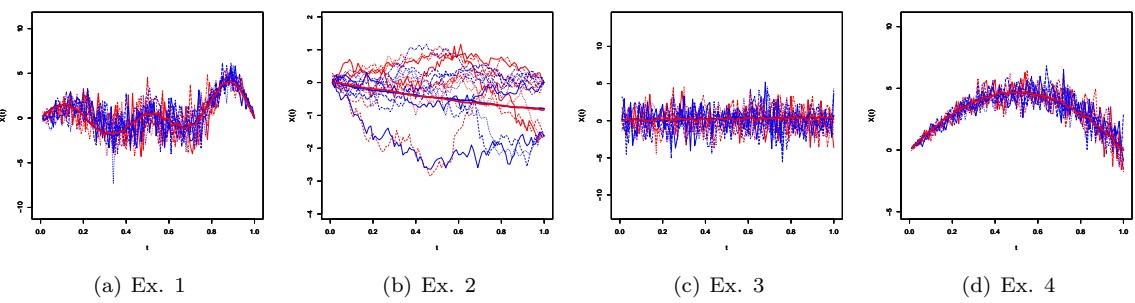

(a) Ex. 1      (b) Ex. 2      (c) Ex. 3      (d) Ex. 4

Figure 2: Representative curves for the four examples having differences in locations and scales. The mean curves marked in bold, while the two classes marked in red and blue colors, respectively. Please zoom in for a better resolution.

In the first setting, we considered clustering problems with differences in their `location and scale` parameters. Class specific representative plots are given in Figure 2. Usefulness of the proposed transformation is clear from Table 4.1. Our method attains the first position across all examples, while in Example IV we

obtain perfect clustering. Although there is no location difference in Example II, sub-optimal performance of our method is probably due to low signal from the difference between the two covariance structures. CL attains the second best performance in the first three examples among the competing methods, while DHP performs better than CL in Example IV.

In the next setting, we dealt with differences only in `scale` parameters. Class specific representative plots are given in Figure 3. It is clear from Table 4.2 that the separation in scatters is captured very well by the proposed transformation $\hat{\Gamma}_d$. Moreover, our method again leads to perfect clustering (with a significant improvement in Example II compared to Table 4.1). In Example II, all (except possibly, funclust) the competing methods perform quite poorly across all examples. However, its performance is far below our method CD even in this case. The performances of $k$-means and DHP are similar, and quite bad in this scenario. Generally, the results in Table 4.2 suggest that all existing methods fail to judiciously capture information if it is present only in the scale parameters.

Table 4.2: Adjusted Rand distances for different GPs with differences only in scales (with standard error in brackets).

| Ex. | $k$-means | spectral | mclust | funclust | CL | DHP | CD | | |
|-----|-----------|----------|--------|----------|-----|-----|--------|----------|--------|
|     |           |          |        |          |     |     | $k$-means | spectral | mclust |
| I   | 0.9900    | 0.9992   | 1.0000 | 0.9776   | 0.8269 | 0.9966 | 0.0063 | *0.0001* | **0.0000** |
|     | (0.0014)  | (0.0007) | (0.0000) | (0.0006) | (0.0000) | (0.0005) | (0.0002) | (0.0001) | (0.0000) |
| II  | 0.9985    | 0.9962   | 0.9986 | 0.5004   | 0.9065 | 0.9999 | *0.0091* | 0.0100 | **0.0084** |
|     | (0.0005)  | (0.0007) | (0.0005) | (0.0049) | (0.0007) | (0.0003) | (0.0036) | (0.0047) | (0.0019) |
| III | 0.9944    | 0.9990   | 1.0000 | 0.9956   | 0.9994 | 0.9967 | *0.1789* | 0.2872 | **0.0352** |
|     | (0.0009)  | (0.0004) | (0.0000) | (0.0000) | (0.0000) | (0.0007) | (0.0026) | (0.0032) | (0.0007) |
| IV  | 0.9927    | 0.9984   | 1.0000 | 1.0006   | 0.8464 | 0.9980 | 0.0102 | *0.0014* | **0.0005** |
|     | (0.0002)  | (0.0005) | (0.0000) | (0.0000) | (0.0000) | (0.0006) | (0.0021) | (0.0013) | (0.0004) |

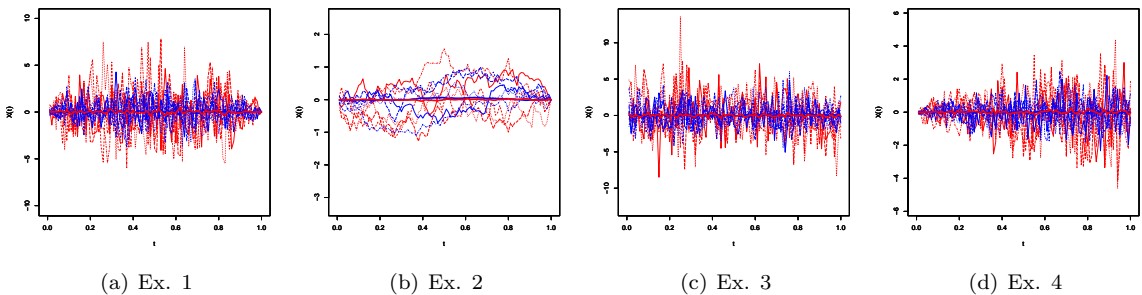

(a) Ex. 1          (b) Ex. 2          (c) Ex. 3          (d) Ex. 4

Figure 3: Representative curves for the four examples having differences only in scales. The mean curves marked in bold, while the two classes marked in red and blue colors, respectively. Please zoom in for a better resolution.

After applying the transformation $\hat{\Gamma}_d$, we had used three methods for clustering the transformed observations (see Tables 4.1 and 4.2). Additionally, when these methods are applied directly to the functional data, we observe that $k$-means and spectral perform fairly well, but their performances deteriorate sharply for the case with differences only in the scales. The mclust algorithm performs worst (possibly due to the presence of low signal in the locations) among the three clustering methods, while CD-mclust clearly achieves the best results with a substantial improvement (see Table 4.1). It is worth mentioning that the Rand distances improve substantially for all three methods when applied on the transformed matrix $\hat{\Gamma}_d$.

All the three usual clustering methods seem useless (possibly due to the presence of signal only in the scales) from the results in Table 4.2, while their CD counterparts clearly lead to excellent performances with CD-mclust yielding the overall best result. Overall, the best performance of this method may be attributed to the fact that the signal gets amplified in the transformed matrix and the specific structure (recall equation (11)) of this matrix.

# 5 Analysis of Benchmark Datasets

We have applied our proposed methods to some benchmark data sets, `Wheat` (from the `R` package `fds`), `Satellite` (available at `https://www.math.univ-toulouse.fr/~ferraty/SOFTWARES/NPFDA/index.html`), `Cars` (kindly provided by the first author of Torrecilla et al (2020)) and `Velib` (from the `R` package `funFEM`).

To evaluate the clustering algorithms, we ran a single execution (without splitting). Class assigments are already available for the `Wheat` dataset. The `Satellite` data has been analyzed in detail in the paper Dabo-Niang et al (2007), where the authors split the curves into two clusters 'unimodal' and 'multimodal'. The authors of this paper kindly shared the exact cluster assignments for this data set with us. The `Cars` data contains asset log-returns of the car companies Tesla, General Motors and BMW (see Torrecilla et al (2020) for more details). However, the rank of the estimated $\hat{\Gamma}_d$ matrix was two for this data set, and our method detected only two distinct clusters. This is coherent with Torrecilla et al (2020), where the authors had noted that assets of General Motors and BMW were very similar and quite difficult to distinguish. So, we merged General Motors with BMW while assigning the class labels for this data set. Consequently, the number of clusters was set to be two for all the competing methods. The `Velib` data was analyzed by Bouveyron et al (2015), where the authors identified the optimal number of clusters to be ten using the `funFEM` algorithm. Setting $J = 10$ (this information was provided to the competing methods), we determined the class labels of the observations using this algorithm. We report the adjusted Rand distance for these four data sets in Table 5.1. Competitive performance of our proposed methodology w.r.t. the competing methods is clear from the results given below.

Table 5.1: Adjusted Rand distances for different clustering methods.

| Data | $J$ | $N$ | $d$ | $k$-means | spectral | mclust | funclust | CL | DHP | CD | | |
|---|---|---|---|---|---|---|---|---|---|---|---|---|
| | | | | | | | | | | $k$-means | spectral | mclust |
| Wheat | 2 | 100 | 701 | 0.6960 | 0.7860 | 0.6978 | 0.6960 | 0.8058 | *0.5730* | 0.7859 | 0.8057 | **0.3644** |
| Satellite | 2 | 472 | 70 | 0.6072 | 0.8947 | **0.3525** | 0.6072 | 0.6060 | 0.7253 | 0.8861 | 0.9954 | *0.4448* |
| Cars | 2 | 90 | 32 | 0.8856 | 0.9230 | 0.8834 | 0.8856 | 0.9650 | 0.9088 | *0.5385* | 0.9313 | **0.4680** |
| Velib | 10 | 1189 | 181 | **0.3755** | 0.5464 | 0.8561 | $*$ | 0.6738 | $**$ | 0.5672 | 0.5872 | *0.5408* |

$*$ - R package is now archived; $**$ - Matlab code available only for two classes.

To get a better understanding of the performance of our proposed method, we further computed the well-known average purity function correspoinding to the CD method having the minimum adjusted Rand distance. A value of average purity function close to one indicates good performance of a method. We obtained the values as 0.9000, 0.8622, 0.8666 and 0.6038 for the `Wheat` data, the `Satellite` data, the `Cars` data and the `Velib` data, respectively. Overall, our proposed method CD yields quite promising results in all four benchmark data sets, with CD-mclust yielding the most stable performance.

# Acknowledgements

The first author has been partially supported by grants PID2021-128314NB-I00 and PID2022-139237NB-I00 funded by MCIN/AEI/10.13039/501100011033/FEDER, UE and "ERDF A way of making Europe". Both the authors would like to thank the Action Editor Prof. Brian Kulis and three anonymous reviewers for their constructive comments and suggestions that improved the paper.

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

## Appendix I: Proofs and Mathematical Details

## A    Proof of Theorem 2.1

Fix $d \in \mathbb{N}$. The $d$-dimensional random vector $(\mathbf{Z} - b)_d$ has a Gaussian distribution with mean equal to $(\mu - b)_d$ and covariance matrix equal to $\Sigma_d^{\mathbf{Z}}$. Now, $\|(A_d)^{-1/2}(\mathbf{Z} - b)_d\|^2$ is equal to the square of the norm of a $d$-dimensional normal variable with mean $\mathbf{m}_d = (A_d)^{-1/2}(\mu - b)_d$ and covariance matrix $S_d = (A_d)^{-1/2}\Sigma_d^{\mathbf{Z}}(A_d)^{-1/2}$. Therefore, if $\mathbf{u}_d$ is a $d$-dimensional vector with centered normal distribution and covariance matrix equal to $S_d$, then

$$D_d^{A_d}(\mathbf{Z}, b) \sim \frac{1}{d}\langle \mathbf{m}_d + \mathbf{u}_d, \mathbf{m}_d + \mathbf{u}_d \rangle = \frac{1}{d}\left( \|\mathbf{m}_d\|^2 + \|\mathbf{u}_d\|^2 + 2\langle \mathbf{m}_d, \mathbf{u}_d \rangle \right). \tag{21}$$

By assumption (3), we have

$$\lim_{d \to \infty} \frac{1}{d}\|\mathbf{m}_d\|^2 = L_\mu.$$

Let us consider the *second term* in (21). Fix a basis in $V_d$ spanned by the eigenvectors of $S_d$. Note that this term is not dependent on $L_\mu$. Denote $\mathbf{u}_d = (u_{d,1}, \ldots, u_{d,d})^T$ and $\mathbf{m}_d = (m_{d,1}, \ldots, m_{d,d})^T$ in this basis. Therefore, the random variables $(u_{d,i})^2$ with $1 \leq i \leq d$ are independent with means equal to $\alpha_i^d$ for $1 \leq i \leq d$ and $\sum_{i=1}^{d}(u_{d,i})^2 \sim \sum_{i=1}^{d}\alpha_i^d(u_i)^2$. Here, $\{u_i\}_{1 \leq i \leq d}$ is a sequence of independent and identically distributed (i.i.d.) real variables with the standard normal distribution. We split the proof into two cases.

### A.0.1    $L_S$ is finite

Fix $\epsilon > 0$. Taking into account that the variance of a $\chi^2$ distribution with one degree of freedom is two and using Chebychev's inequality, we have that

$$
\begin{aligned}
\mathbb{P}\left[ \frac{1}{d}\left| \|\mathbf{u}_d\|^2 - \operatorname{trace}(S_d) \right| \geq \epsilon \right] &= \mathbb{P}\left[ \frac{1}{d}\left| \sum_{i=1}^{d}\left( (u_{d,i})^2 - \alpha_i^d \right) \right| \geq \epsilon \right] \\
&\leq \frac{2}{\epsilon^2 d^2}\sum_{i=1}^{d}(\alpha_i^d)^2 \\
&\leq \frac{2}{\epsilon^2 d^2}\|\alpha^d\|_\infty \sum_{i=1}^{d}\alpha_i^d,
\end{aligned}
$$

which converges to zero by assumptions (4) and (2). Consequently, we have shown that

$$\frac{1}{d}\|\mathbf{u}_d\|^2 - \frac{1}{d}\operatorname{trace}(S_d) \xrightarrow{P} 0 \text{ as } d \to \infty,$$

and assumption (4) gives

$$\frac{1}{d}\|\mathbf{u}_d\|^2 \xrightarrow{P} L_S \text{ as } d \to \infty.$$

### A.0.2    $L_S$ is infinite

We have that

$$
\begin{aligned}
\mathbb{P}\left[ \frac{1}{\sum_{i=1}^{d}\alpha_i^d}\left| \sum_{i=1}^{d}\left( (u_{d,i})^2 - \alpha_i^d \right) \right| \geq \epsilon \right] &= \mathbb{P}\left[ \left| \sum_{i=1}^{d}\frac{\alpha_i^d}{\sum_{i=1}^{d}\alpha_i^d}\left( (u_i)^2 - 1 \right) \right| \geq \epsilon \right] \\
&\leq \frac{2}{\epsilon^2}\sum_{i=1}^{d}\left( \frac{\alpha_i^d}{\sum_{i=1}^{d}\alpha_i^d} \right)^2 \\
&\leq \frac{2}{\epsilon^2}\frac{\|\alpha^d\|_\infty}{\sum_{i=1}^{d}\alpha_i^d},
\end{aligned}
$$

which converges to zero because $L_S = \infty$ and assumption (2). Thus, we have shown that

$$\frac{1}{\frac{1}{d}\sum_{i=1}^{d}\alpha_i^d}\left(\frac{1}{d}\|\mathbf{u}_d\|^2 - \frac{1}{d}\text{trace}(S_d)\right) \xrightarrow{P} 0 \text{ as } d \to \infty. \tag{22}$$

Consequently, $\frac{1}{d}\|\mathbf{u}_d\|^2$ converges to $\infty$ at the same rate as $\frac{1}{d}\text{trace}(S_d)$.

Concerning the *last term* in (21), we have $\langle\mathbf{m}_d, \mathbf{u}_d\rangle = \sum_{i=1}^{d} m_{d,i}u_{d,i}$. We split the proof into cases.

### A.0.3 $L_\mu$ is finite

Fix $\epsilon > 0$, and define $\alpha^d = (\alpha_1^d, \ldots, \alpha_d^d)^T$. Using Chebychev's inequality again, we get

$$\mathbb{P}\left[\frac{1}{d}|\langle\mathbf{m}_d, \mathbf{u}_d\rangle| > \epsilon\right] \leq \frac{1}{\epsilon^2 d^2}\sum_{i=1}^{d}(m_{d,i})^2\alpha_i^d \leq \frac{1}{\epsilon^2 d^2}\|\alpha^d\|_\infty\|\mathbf{m}_d\|^2,$$

which converges to zero by assumptions (3) and (2), and the proposition is proved in this case.

### A.0.4 $L_\mu$ is infinite

The result follows from equation (21) and the previous results, if we are able to show that the sequence of real-valued random variables
$$w_d = \frac{\langle\mathbf{m}_d, \mathbf{u}_d\rangle}{\max(\|\mathbf{m}_d\|^2, \|\mathbf{u}_d\|^2)}$$
converges to zero in probability as $d \to \infty$. In turn, this will be fixed if we show that every subsequence of $\{w_d\}$ contains a new subsequence which satisfies this property. Thus, let $\{w_{d_k}\}$ be a subsequence of $\{w_d\}$ and let us consider the associated subsequences $\{\|\mathbf{m}_{d_k}\|\}$ and $\{\|\mathbf{u}_{d_k}\|\}$. Obviously, there exists a further subsequence $\{d_{k^*}\}$ such that one of the following holds:

(i) $\lim_{d_{k^*}}\dfrac{\|\mathbf{m}_{d_{k^*}}\|^2}{\text{trace}(S_{d_{k^*}})} = 0$.

(ii) $\lim_{d_{k^*}}\dfrac{\|\mathbf{m}_{d_k}\|^2}{\text{trace}(S_{d_{k^*}})} = \infty$.

(iii) There exists a finite $C > 0$ such that $\lim_{d_{k^*}}\dfrac{\|\mathbf{m}_{d_{k^*}}\|^2}{\text{trace}(S_{d_{k^*}})} = C$.

Note that in cases (i) and (iii), we have $L_S = \infty$. To simplify notation, we denote the sequence $\{S_{d_{k^*}}\}$ by $\{S_h\}$, and similarly for the remaining ones. In case (i), since equation (22) shows that

$$\frac{\|\mathbf{u}_h\|^2}{\text{trace}(S_h)} \xrightarrow{P} 1 \text{ as } h \to \infty, \tag{23}$$

we have $\dfrac{\|\mathbf{m}_h\|}{\|\mathbf{u}_h\|} \xrightarrow{P} 0$ as $h \to \infty$. Consequently,

$$\lim_{h}|w_h| = \lim_{h}\frac{|\langle\mathbf{m}_h, \mathbf{u}_h\rangle|}{\|\mathbf{u}_h\|^2} \leq \lim_{h}\frac{\|\mathbf{m}_h\|}{\|\mathbf{u}_h\|} = 0 \text{ in probability.}$$

If (ii) holds, we have that $|w_h| \leq \frac{\|\mathbf{u}_h\|}{\|\mathbf{m}_h\|}$. Since $E\left[\|\mathbf{u}_h\|^2\right] = \text{trace}(S_d)$, we have that $\frac{\|\mathbf{u}_h\|^2}{\|\mathbf{m}_h\|^2} \xrightarrow{P} 0$ and also, in this case $w_h \xrightarrow{P} 0$ as $h \to \infty$.

In case (iii), taking into account that equation (23) now holds, it is enough to show that

$$\frac{\langle\mathbf{m}_h, \mathbf{u}_h\rangle}{C\text{trace}(S_h)} \xrightarrow{P} 0 \text{ as } h \to \infty,$$

Fix $\epsilon > 0$. We have that

$$\mathbb{P}\left[\left|\frac{\langle \mathbf{m}_h, \mathbf{u}_h \rangle}{C\,\mathrm{trace}(S_h)}\right| > \epsilon\right] \quad \leq \quad \frac{1}{C^2\epsilon^2}\sum_{i=1}^{h}\frac{m_{h,i}^2\alpha_i^h}{\left(\sum_{i=1}^{h}\alpha_i^h\right)^2} \leq \frac{1}{C^2\epsilon^2}\frac{\|\alpha^h\|_\infty}{\sum_{i=1}^{h}\alpha_i^h}\frac{\|\mathbf{m}_h\|^2}{\mathrm{trace}(S_h)},$$

which converges to zero by assumptions (4) and (2). ●

## B  On Assumptions (4) and (2)

The next lemma shows that if $L_S < \infty$, then assumption (4) implies assumption (2).

**Lemma B.1** *Let $\{a_d\}_{d\geq 1}$ be a sequence of real positive numbers such that $\lim_d \frac{1}{d}\sum_{i=1}^{d} a_i$ exists, and it is finite. Then, it happens that $\lim_d \frac{1}{d}\|a^d\|_\infty = 0$.*

*Proof:* Fix $d \in \mathbb{N}$, and denote $A_d = \sum_{i=1}^{d} a_i$. We have that

$$\frac{a_d}{d} = \frac{A_d}{d} - \frac{A_{d-1}}{d-1}\frac{d-1}{d}$$

and consequently, $0 = \lim_d \frac{a_d}{d}$. Given $\epsilon > 0$, there exists $d_1 > 0$ such that if $d > d_1$, then $\frac{a_d}{d} \leq \epsilon$ and $d_2 \geq d_1$ such that

$$\sup_{1\leq i\leq d_1}\frac{a_i}{d_2} \leq \epsilon.$$

Let $d > d_2$ and take $1 \leq i \leq d$. So, we have that if $i \leq d_1$, then $\frac{a_i}{d} < \frac{a_i}{d_2} \leq \epsilon$ and if $i > d_1$, then $\frac{a_i}{d} \leq \frac{a_i}{i} \leq \epsilon$. This completes the proof. ●

## C  Proof of Theorem 2.2

First, note that $(\mathbf{Z}_1 - \mathbf{Z}_2)_d$ is a $d$-dimensional normal vector, with mean $(\mu^{\mathbf{Z}_h} - \mu^{\mathbf{Z}_k})_d$ and covariance $\Sigma_d^{\mathbf{Z}_h} + \Sigma_d^{\mathbf{Z}_k}$ for $h, k \in \{1, 2\}$. Now, the proof will based in the following lemma.

**Lemma C.1** *Assume that $h \neq k \in \{1, 2\}$. Define $S_d^h, S_d^k$ and $S_d^{hk}$ as in Theorem 2.2. Then,*

*i) $L_\mu^{hk} = \lim_d D_d^{\Sigma_d^{\mathbf{Z}}}(\mu^{\mathbf{Z}_h}, \mu^{\mathbf{Z}_k}) = 0$, and*

*ii) $\limsup_d \frac{1}{d}trace(S_d^h) < \infty$, $\limsup_d \frac{1}{d}trace(S_d^k) < \infty$ and $\limsup_d \frac{1}{d}trace(S_d^{hk}) < \infty$.*

*Proof:* Let us denote $\Sigma^* = \pi_1\Sigma^{\mathbf{Z}_h} + \pi_2\Sigma^{\mathbf{Z}_k}$, $\mu = (\mu^{\mathbf{Z}_h} - \mu^{\mathbf{Z}_k})$ and $\pi_{12} = \pi_1\pi_2$. From (5), we have that

$$\Sigma_d^{\mathbf{Z}} = \Sigma_d^* + \pi_{12}\mu_d\mu_d^T.$$

From here, the Sherman-Morrison formula gives

$$(\Sigma_d^{\mathbf{Z}})^{-1} = (\Sigma_d^*)^{-1} - \frac{\pi_{12}(\Sigma_d^*)^{-1}\mu_d\mu_d^T(\Sigma_d^*)^{-1}}{1 + \pi_{12}\mu_d^T(\Sigma_d^*)^{-1}\mu_d}.$$

Since $(\Sigma_d^*)^{-1}$ is positive definite (p.d.) for all $d \in \mathbb{N}$, this now implies that

$$0 \leq \mu_d^T(\Sigma_d^{\mathbf{Z}})^{-1}\mu_d = \mu_d^T(\Sigma_d^*)^{-1}\mu_d - \frac{\pi_{12}(\mu_d^T(\Sigma_d^*)^{-1}\mu_d)^2}{1 + \pi_{12}(\mu_d^T(\Sigma_d^*)^{-1}\mu_d)} = \frac{\mu_d^T(\Sigma_d^*)^{-1}\mu_d}{1 + \pi_{12}\mu_d^T(\Sigma_d^*)^{-1}\mu_d} \leq \frac{1}{\pi_{12}},$$

and the proof that $L_\mu^{hk} = 0$ trivially ends from definition of $L_\mu^{hk}$.

To handle the terms $L_S^h$ ($L_S^k$ is identical) and $L_S^{hk}$, recall the Woodbury matrix identity:

$$(U + V)^{-1} = U^{-1} - (U + UV^{-1}U)^{-1}.$$

Using this identity, we have

$$(\Sigma_d^{\mathbf{Z}})^{-1} = 1/\pi_1(\Sigma_d^{\mathbf{Z}_h})^{-1} - B_d,$$

where $B_d = (\pi_1\Sigma_d^{\mathbf{Z}_h} + \pi_1^2\Sigma_d^{\mathbf{Z}_h}(\pi_2\Sigma_d^{\mathbf{Z}_k} + \pi_{12}\mu_d\mu_d^T)^{-1}\Sigma_d^{\mathbf{Z}_h})^{-1}$.

If $U$ and $V$ are p.d., then $U^TVU$ is p.d. In $B_d$, both the matrices $\Sigma_d^{\mathbf{Z}_h}$ and $(\pi_2\Sigma_d^{\mathbf{Z}_k} + \pi_{12}\mu_d\mu_d^T)$ are symmetric and p.d., and this implies that $B_d$ is also p.d. Further, $(\Sigma_d^{\mathbf{Z}_h})^{1/2}$ and $B_d$ are p.d. which now implies that $(\Sigma_d^{\mathbf{Z}_h})^{1/2}B_d(\Sigma_d^{\mathbf{Z}_h})^{1/2}$ is p.d. Recall that *trace* is a linear map. Now,

$$\begin{aligned}
trace(S_d^h) &= 2\,trace(\Sigma_d^{\mathbf{Z}_h}(\Sigma_d^{\mathbf{Z}})^{-1}) \\
&= 2\,trace(1/\pi_1 I_d) - trace(\Sigma_d^{\mathbf{Z}_h}B_d) \\
&= 2\,trace(1/\pi_1 I_d) - trace((\Sigma_d^{\mathbf{Z}_h})^{1/2}B_d(\Sigma_d^{\mathbf{Z}_h})^{1/2}) \\
&\leq trace(2/\pi_1 I_d) = 2d/\pi_1.
\end{aligned}$$

We have proved that $\limsup_d \frac{1}{d}trace(S_d^h) < 2/\pi_1$. Similarly, it is proved that $\limsup_d \frac{1}{d}trace(S_d^k) < 2/\pi_2$. Finally, we have

$$\begin{aligned}
trace(S_d^{hk}) &= trace((\Sigma_d^{\mathbf{Z}})^{-1/2}\Sigma_d^{\mathbf{Z}_h}(\Sigma_d^{\mathbf{Z}})^{-1/2}) + trace((\Sigma_d^{\mathbf{Z}})^{-1/2}\Sigma_d^{\mathbf{Z}_k}(\Sigma_d^{\mathbf{Z}})^{-1/2}) \\
&\leq \frac{d}{\pi_1} + \frac{d}{\pi_2} = \frac{d}{\pi_1\pi_2}
\end{aligned}$$

from where we conclude that $\limsup_d \frac{1}{d}trace(S_d^{hk}) < 1/(\pi_1\pi_2)$. ●

To prove part (*a*) in Theorem 2.2, we have $h = k$. So, $(\mu^{\mathbf{Z}_h} - \mu^{\mathbf{Z}_k})_d = 0_d$ and $\Sigma_d^{\mathbf{Z}_h} + \Sigma_d^{\mathbf{Z}_k} = 2\Sigma_d^{\mathbf{Z}_h}$. If we take $A_d = S_d^h$, according to Remark 2.1.2, Lemma C.1 gives that assumption (2) holds for this selection of $A_d$. Therefore, (7) follows from Theorem 2.1 because in this case $L_\mu^h = 0$ and once we assume that $L_S^h$ exists, Lemma C.1 gives that it is finite.

In case (b), we have $h \neq k$. We take $A_d = S_d^{hk}$ and $b = \mu^{\mathbf{Z}_h} - \mu^{\mathbf{Z}_k}$. Similarly, as in (a), we have that assumption (2) also holds in this case and Theorem 2.1 implies

$$D_d^{\Sigma_d^{\mathbf{Z}}}(\mathbf{Z}_1, \mathbf{Z}_2) \xrightarrow{P} L_\mu^{hk} + L_S^{hk} \text{ as } d \to \infty.$$

Now, (8) follows because Lemma C.1 gives that $L_\mu^{hk} = 0$ and, also, that $L_S^{hk}$ is finite. ●

## D  Proof of Proposition 2.3

Under the conditions of Proposition 2.3, the number of significant (unique) eigenvalues of the matrix $\Gamma$ is 2. Recall that $N$ is fixed here.

Consider the standardized distance matrix $D_d$ with the $(i, j)$-th element as $D_d^{\Sigma_d^{\mathbf{Z}}}(\mathbf{Z}_i, \mathbf{Z}_j)$ for $1 \leq i, j \leq N$ and $d \in \mathbb{N}$. We have a sequence of matrices $D_d \xrightarrow{P} D_0$ as $d \to \infty$ (componentwise). Since the map $D$ to $\Gamma$ is clearly continuous w.r.t. this convergence, we have that $\Gamma_d \xrightarrow{P} \Gamma$ as $d \to \infty$. Let us denote the eigenvalues of $\Gamma_d$ (respectively, $\Gamma$) to be $\beta_1^d, \ldots, \beta_N^d$ (respectively, $\beta_1, \ldots, \beta_N$). Since eigenvalues are continuous functions of the respective matrices, we have $\beta_j^d \xrightarrow{P} \beta_j$ as $d \to \infty$ for all $1 \leq j \leq N$.

Let us now look into the following:

$$\sum_{i=1}^{N} I(|\beta_i^d| > a_d) \xrightarrow{P} \sum_{i=1}^{N} I(|\beta_i^0| > 0) \text{ as } d \to \infty$$

with $a_d \downarrow 0$ as $d \to \infty$ at an appropriate rate. Recall that the limiting quantity on the right should give us the correct number of clusters. Consider the sequence $\{1/m\}_{m \in \mathbb{N}}$. Let us take $i \in \{1, \ldots, N\}$ such that $\beta_i^d \xrightarrow{P} 0$ as $d \to \infty$. Thus, for every $\epsilon, \delta > 0$ there exists $D_{\delta, \epsilon}^i$ such that if $d \geq D_{\delta, \epsilon}^i$ then

$$\mathbb{P}[|\beta_i^d| > \delta] < \epsilon.$$

In particular, if we take $\delta = \epsilon = 1/m$, there exists $D_m^i$ such that if $d \geq D_m^i$, then we have

$$\mathbb{P}\left[|\beta_i^d| > \frac{1}{m}\right] < \frac{1}{m}.$$

Without loss of generality, we can assume that $D_1^i < D_2^i < \cdots$, and consider the sequence

$$a_d^i = \begin{cases} 2 & \text{if } 1 \leq i < D_1^i, \\ \frac{1}{m} & \text{if } D_m^i \leq i < D_{m+1}^i \text{ for some } m \geq 1. \end{cases}$$

Then, obviously $a_d^i \to 0$ and

$$\mathbb{P}\left[I(|\beta_i^d| > a_d^i) > 0\right] = \mathbb{P}\left[|\beta_i^d| > a_d^i\right] < a_d^i.$$

If we define $a_d = \sup\{a_d^i : \beta_i^0 = 0\}$, and $i$ satisfies that $\beta_i^0 = 0$, then $I(|\beta_i^d| > a_d) \xrightarrow{P} 0$ as $d \to \infty$. A similar reasoning allows us also to conclude that if $|\beta_i^0| > 0$, then $I(|\beta_i^d| > a_d) \xrightarrow{P} 1$ as $d \to \infty$. $\qquad \bullet$

## E   Proof of Theorem 2.5

In this proof, we use the superindex $d$ in $\mathcal{G}_i^d$ to emphasize that the groupings can change with the dimension $d \in \mathbb{N}$. Proposition 2.3 implies that $K_d = 2$ with probability converging to one.

Note that $\phi(\mathcal{G}_1, \ldots, \mathcal{G}_J)$ has an alternative mathematical expression as

$$\sum_{h=1}^{J} \frac{1}{2|\mathcal{G}_h|} \sum_{\mathbf{u}, \mathbf{v} \in \mathcal{G}_h} \|\mathbf{u} - \mathbf{v}\|^2, \tag{24}$$

where $|\mathcal{G}|$ denotes the cardinality of the set $\mathcal{G}$. Let us denote the rows/columns of $\Gamma_d$ as $\boldsymbol{\gamma}_1^d, \ldots, \boldsymbol{\gamma}_N^d$. The structure of $\Gamma^d$ implies that $\|\boldsymbol{\gamma}_i^d - \boldsymbol{\gamma}_j^d\|^2 \xrightarrow{P} 0$ as $d \to \infty$ iff $i, j \in \mathcal{C}_h$ for $h \in \{1, 2\}$. So, if each $\mathcal{G}_h^d$ for $h = 1, 2$ contains observations from the same population, then $\phi_d(\mathcal{G}_1^d, \mathcal{G}_2^d) \xrightarrow{P} 0$ as $d \to \infty$.

Let us assume that on the contrary, there exists a subsequence of dimensions $\{d_k\}$ such that for every $k$ there exists at least a couple of points $i_k, j_k$ with $i_k \in \mathcal{G}_1^d$ and $j_k \in \mathcal{G}_2^d$ (say). Since the number of points is finite, there exists a further subsequence $\{d_{k^*}\}$ such that both sequences $\{i_{k^*}\}$ and $\{j_{k^*}\}$ are constant. Therefore, for those subsequences, we will have

$$\liminf_d \phi_d(\mathcal{G}_1, \mathcal{G}_2) \geq \lim_d \|\boldsymbol{\gamma}_{i_{k^*}}^{d_{k^*}} - \boldsymbol{\gamma}_{j_{k^*}}^{d_{k^*}}\|^2 \xrightarrow{P} \gamma_{12} > 0.$$

So, for the minimization of $\phi_d(\mathcal{G}_1^d, \mathcal{G}_2^d)$, each $\mathcal{G}_h^d$ must contain all observations from a single population with probability converging to one as the dimension increases. This proves the convergence in probability of the Rand index $\mathbb{R}_{d,N}$ to zero as $d \to \infty$. $\qquad \bullet$

## F   Rank of the Matrix $\Gamma$

Identifying number of clusters from the matrix $\Gamma$ is not equivalent to finding the rank of the matrix $\Gamma$.

**Lemma F.1** *The rank of the matrix $\Gamma$ is less then or equal to $J$. Moreover, equality is guaranteed only when $J \leq 3$.*

*Proof:* Trivially, $rank(\Gamma) \leq J$. Let us denote the reduced Echelon form of $N \times N$ matrix $\Gamma$ as $\Gamma^\circ$. Thus, the matrix $\Gamma^\circ$ is a $J \times J$ symmetric matric with $\gamma_{ij} > 0$ and distinct when $i \neq j$, while $\gamma_{ii} = 0$.

Moreover, for $J = 3$, we have

$$det(\Gamma^\circ) = det \begin{pmatrix} 0 & \gamma_{12} & \gamma_{13} \\ \gamma_{12} & 0 & \gamma_{23} \\ \gamma_{13} & \gamma_{23} & 0 \end{pmatrix} = 2\gamma_{12}\gamma_{13}\gamma_{23} \neq 0.$$

In the case $J = 4$, if $\gamma_{12} = \dfrac{\gamma_{13}\gamma_{24} + \gamma_{14}\gamma_{23} + 2\sqrt{\gamma_{13}\gamma_{14}\gamma_{23}\gamma_{24}}}{\gamma_{34}}$, then a simple computation gives that $det(\Gamma^\circ) = 0$. This happens, for instance, if we consider the following matrix (with all *positive and distinct* off-diagonal entries):

$$\begin{pmatrix} 0 & t & 1 & 2 \\ t & 0 & 3 & 4 \\ 1 & 3 & 0 & 5 \\ 2 & 4 & 5 & 0 \end{pmatrix},$$

where $t = 2 + 4\sqrt{6}/5 > 0$. $\bullet$

## G  Proof of Theorem 2.6

In order to simplify the writing, we will write $d$ instead of $d_N$. We will use the notation $\|\alpha_d\|_2 := \left(\sum_{i=1}^d (\alpha_i^d)^2\right)^{1/2}$. The real r.v.'s $\{u_i\}$ are assumed to be i.i.d. with standard normal distribution.

The following lemma is deduced from Lemma 1 in Laurent and Massart (2000) on p. 1325, after some simple computations, taking into account that $\|\alpha_d\|_2 \geq \|\alpha_d\|_\infty$. We state it here for further reference.

**Lemma G.1** *If $Z_d = \sum_{i=1}^d \alpha_i^d(u_i^2 - 1)$ and $x \geq 1$, then*

$$\mathbb{P}\left[|Z_d| \geq 4x\|\alpha_d\|_\infty\right] \leq 2\exp(-x).$$

We will also employ the following well known bound for the tail of the standard normal distribution:

$$\mathbb{P}[|N(0,1)| \geq t] \leq \sqrt{\frac{2}{\pi}} \exp(-t^2/2) \text{ for all } t \geq 1. \tag{25}$$

*Proof of Theorem 2.6*: Let us show part b). The proof of (14) is similar to that of (16). We use the notation $\mathbf{m}_d = (\Sigma_d^{\mathbf{Z}})^{-1/2}(\mu^{\mathbf{Z}_1} - \mu^{\mathbf{Z}_2})_d$ and $\mathbf{u}_d^i = (\Sigma_d^{\mathbf{Z}})^{-1/2}(\mathbf{Z}_i - \mu_i)_d$ with $d \in \mathbb{N}$, where $\mathbf{Z}_i$ is a generic observation with distribution $\mathbb{P}_i$ for $i = 1, 2$. Moreover, with an obvious abuse of notation, we will often write $\mathbf{u}_d^i \in \mathcal{C}_i^N$ with $d \in \mathbb{N}$ for $i = 1, 2$.

Recall that $L_\mu = 0$ and $L_S < \infty$ (see part (c) in Theorem 2.2). Repeating the first steps in the proof of Theorem 2.1, we have that

$$\sup_{\mathbf{Z}^1 \in \mathcal{C}_1^N, \mathbf{Z}^2 \in \mathcal{C}_2^N} \left| D_d^{\Sigma_d^{\mathbf{Z}}}(\mathbf{Z}_1, \mathbf{Z}_2) - \frac{1}{d}\text{trace}(S_d^{12}) \right|$$

$$\leq \left| \frac{1}{d}\|\mathbf{m}_d\|^2 \right| + \sup_{\mathbf{u}^1 \in \mathcal{C}_1^N, \mathbf{u}^2 \in \mathcal{C}_2^N} \left| \frac{1}{d}\|\mathbf{u}_d^1 - \mathbf{u}_d^2\|^2 - \frac{1}{d}\text{trace}(S_d^{12}) \right| \tag{26}$$

$$+ 2\sup_{\mathbf{u}^1 \in \mathcal{C}_1^N, \mathbf{u}^2 \in \mathcal{C}_2^N} \frac{1}{d}\left| \langle \mathbf{m}_d, \mathbf{u}_d^1 - \mathbf{u}_d^2 \rangle \right|, \tag{27}$$

and it is enough to prove that the terms in (26) and (27) converge to zero in probability.

The `first term` in (26) converges to zero by first part of (c) in Theorem 2.2. Concerning the `second term`, let $N_1, N_2$ be the number of elements in $\mathcal{C}_1^N$ and $\mathcal{C}_2^N$, respectively. Since $N_1 + N_2 = N$, it is clear that $N_1 N_2 \leq N^2/4$. Let $\varepsilon > 0$. We have that

$$
\begin{aligned}
P_N \quad &:= \quad \mathbb{P}\left[ \sup_{\mathbf{u^1} \in \mathcal{C}_1^N, \mathbf{u^2} \in \mathcal{C}_2^N} \left| \frac{1}{d}\|\mathbf{u}_d^1 - \mathbf{u}_d^2\|^2 - \frac{1}{d}\mathrm{trace}(S_d^{12}) \right| > \varepsilon \right] \\
&= \quad \mathbb{P}\left[ \bigcup_{\mathbf{u^1} \in \mathcal{C}_1^N, \mathbf{u^2} \in \mathcal{C}_2^N} \left\{ \left| \frac{1}{d}\|\mathbf{u}_d^1 - \mathbf{u}_d^2\|^2 - \frac{1}{d}\mathrm{trace}(S_d^{12}) \right| > \varepsilon \right\} \right] \\
&\leq \quad \frac{N^2}{4}\mathbb{P}\left[ \left| \frac{1}{d}\|\mathbf{u}_d^1 - \mathbf{u}_d^2\|^2 - \frac{1}{d}\mathrm{trace}(S_d^{12}) \right| > \varepsilon \right],
\end{aligned}
\tag{28}
$$

where $\mathbf{u}^1$ and $\mathbf{u}^2$ are associated with some $\mathbf{Z}_1 \in \mathcal{C}_1^N$ and $\mathbf{Z}_2 \in \mathcal{C}_2^N$, respectively. However, it is clear that

$$
\frac{1}{d}\|\mathbf{u}_d^1 - \mathbf{u}_d^2\|^2 - \frac{1}{d}\mathrm{trace}(S_d^{12}) \sim \frac{1}{d}\sum_{i=1}^d \alpha_i^d(u_i^2 - 1).
$$

Take $x = \varepsilon d/(4\|\alpha_d\|_\infty)$. By assumption (15), we have $d/\|\alpha_d\|_\infty \to \infty$ and eventually $x \geq 1$. So, from Lemma G.1, we obtain

$$
P_N \leq \frac{N^2}{4}\mathbb{P}\left[ \left| \sum_{i \leq d} \alpha_i^d(u_i^2 - 1) \right| > \varepsilon d \right] \leq \frac{1}{2}\exp\left( -\frac{\varepsilon d}{4\|\alpha^d\|_\infty} + 2\log N \right),
$$

which converges to zero by assumption (15).

Fix $\epsilon > 0$. For the `third term`, in equation (27) we have that

$$
\begin{aligned}
P_N^* \quad &:= \quad \mathbb{P}\left[ \sup_{\mathbf{u^1} \in \mathcal{C}_1^N, \mathbf{u^2} \in \mathcal{C}_2^N} \frac{1}{d}|\langle \mathbf{m}_d, \mathbf{u}_d^1 - \mathbf{u}_d^2 \rangle| > \varepsilon \right] \\
&\leq \quad \frac{N^2}{4}\mathbb{P}\left[ \frac{1}{d}|\langle \mathbf{m}_d, \mathbf{u}_d^1 - \mathbf{u}_d^2 \rangle| > \varepsilon \right] \\
&= \quad \frac{N^2}{4}\mathbb{P}\left[ \frac{1}{d}\left| \sum_{i \leq d} m_{di}(\alpha_i^d)^{1/2} u_i \right| > \varepsilon \right] \\
&= \quad \frac{N^2}{4}\mathbb{P}\left[ |N(0,1)| > \varepsilon\frac{d}{\sqrt{\sum_{i \leq d}(m_{di})^2 \alpha_i^d}} \right] \\
&\leq \quad \frac{1}{2^{3/2}\pi^{1/2}}\exp\left( -\frac{\varepsilon^2}{2}\frac{d^2}{\sum_{i \leq d}(m_{di})^2 \alpha_i^d} + 2\log N \right) \\
&\leq \quad \frac{1}{2^{3/2}\pi^{1/2}}\exp\left( -\frac{\varepsilon^2}{2}\frac{d^2}{\|\alpha^d\|_\infty \sum_{i \leq d}(m_{di})^2} + 2\log N \right),
\end{aligned}
\tag{29}
$$

which converges to 0 because of the fact that $L_\mu = 0$ (see part (c) in Theorem 2.2) and (15). The same assumption allows us to apply inequality (25) to equation (29). ●

## H Result Related to Remark 2.6.1

**Proposition H.1** *Under assumptions of Theorem 2.2, if we assume that $\frac{\log N}{d_N} \to 0$, then conditions (13) and (15) hold.*

*Proof:* Fix $h \in \{1, 2\}$, and note that

$$(\Sigma_d^{\mathbf{Z}})^{-1} = (\pi_h \Sigma_d^{\mathbf{Z}_h})^{-1} - P_d,$$

where $P_d = (\pi_h \Sigma_d^{\mathbf{Z}_h} + \pi_h \Sigma_d^{\mathbf{Z}_h} (T_d^k)^{-1} \pi_h \Sigma_d^{\mathbf{Z}_h})^{-1}$ is a p.d. matrix and $T_d^k$ is the matrix $\pi_k \Sigma_d^{\mathbf{Z}_k} + \pi_1 \pi_2 [\mu_d^{\mathbf{Z}_1} - \mu_d^{\mathbf{Z}_2}][\mu_d^{\mathbf{Z}_1} - \mu_d^{\mathbf{Z}_2}]^T$ with $k \neq h \in \{1, 2\}$. Further,

$$I_d + \Sigma_d^{\mathbf{Z}} P_d = \frac{1}{\pi_h} \Sigma_d^{\mathbf{Z}} (\Sigma_d^{\mathbf{Z}_h})^{-1}.$$

From here, Weyl's inequality gives

$$1 \leq \alpha_{min}(\frac{1}{\pi_h} \Sigma_d^{\mathbf{Z}} (\Sigma_d^{\mathbf{Z}_h})^{-1}) = \frac{1}{\pi_h} \alpha_{min}(\Sigma_d^{\mathbf{Z}} (\Sigma_d^{\mathbf{Z}_h})^{-1}). \tag{30}$$

Note the fact that the eigenvalues of the matrices $AB$ and $BA$ are same. So, the matrices $S_d^h$ and $\Sigma_d^{\mathbf{Z}_h} (\Sigma_d^{\mathbf{Z}})^{-1}$ will have the same eigenvalues. Furthermore, the eigenvalues of $S_d^h$ are the inverses of the eigenvalues of $\Sigma_d^{\mathbf{Z}} (\Sigma_d^{\mathbf{Z}_h})^{-1}$. Thus, (30) gives that

$$\alpha_{max}(S_d^h) < \frac{2}{\pi_h} \text{(free of } d). \tag{31}$$

We now have

$$\log N = o\left(\frac{d_N}{\alpha_1^{d_N}}\right) \Leftrightarrow \frac{\alpha_1^{d_N} \log N}{d_N} \to 0 \text{ as } N \to \infty.$$

Equation (31) now implies that condition (13) holds if we assume $\frac{\log N}{d_N} \to 0$ as $N \to \infty$.

Fix $h \neq k \in \{1, 2\}$. Our second matrix of interest is

$$S_d^{hk} = (\Sigma_d^{\mathbf{Z}})^{-1/2} (\Sigma_d^{\mathbf{Z}_h} + \Sigma_d^{\mathbf{Z}_k}) (\Sigma_d^{\mathbf{Z}})^{-1/2}.$$

Since the matrices are symmetric, we have

$$\alpha_{max}(S_d^{hk}) \leq \alpha_{max}((\Sigma_d^{\mathbf{Z}})^{-1/2} \Sigma_d^{\mathbf{Z}_h} (\Sigma_d^{\mathbf{Z}})^{-1/2}) + \alpha_{max}((\Sigma_d^{\mathbf{Z}})^{-1/2} \Sigma_d^{\mathbf{Z}_k} (\Sigma_d^{\mathbf{Z}})^{-1/2}).$$

Again, the eigenvalues of $(\Sigma_d^{\mathbf{Z}})^{-1/2} \Sigma_d^{\mathbf{Z}_i} (\Sigma_d^{\mathbf{Z}})^{-1/2}$ and of $\Sigma_d^{\mathbf{Z}_i} (\Sigma_d^{\mathbf{Z}})^{-1}$ will be equal for $i = h, k$. So,

$$
\begin{aligned}
\alpha_{max}(S_d^{hk}) &\leq \alpha_{max}(\Sigma_d^{\mathbf{Z}_h} (\Sigma_d^{\mathbf{Z}})^{-1}) + \alpha_{max}(\Sigma_d^{\mathbf{Z}_k} (\Sigma_d^{\mathbf{Z}})^{-1}) \\
&= \frac{1}{\alpha_{min}(\Sigma_d^{\mathbf{Z}} (\Sigma_d^{\mathbf{Z}_h})^{-1})} + \frac{1}{\alpha_{min}(\Sigma_d^{\mathbf{Z}} (\Sigma_d^{\mathbf{Z}_k})^{-1})} \\
&\leq \frac{1}{\pi_h} + \frac{1}{\pi_k} = \frac{1}{\pi_h \pi_k} \text{ (using equation (30)).}
\end{aligned}
$$

From here, similarly as before, we would obtain that $\frac{\log N}{d_N} \to 0$ implies (15) holds. $\bullet$

## I   Proof of Theorem 3.1

Recall that we use subspaces $V_d$ generated by estimates of the first $d$ eigenfunctions of the covariance of $\mathbf{Z}$.

We begin with some notation and preliminary results which have been taken from Delaigle and Hall (2012) and Hall and Hosseini-Nasab (2006), or follow directly from the results there. Then, we will give the proof of Theorem 3.1. For every $n \in \mathbb{N}$, let us consider

$$
\begin{aligned}
\hat{\Delta}_{\mathbf{Z}}^2 &= \int_0^1 \int_0^1 (\hat{\Sigma}^{\mathbf{Z}}(s, t) - \Sigma^{\mathbf{Z}}(s, t))^2 ds dt, \\
\delta_j^{\mathbf{Z}} &= \min_{k \leq j} (\lambda_k - \lambda_{k+1}).
\end{aligned}
$$

In Delaigle and Hall (2012) and Hall and Hosseini-Nasab (2006), it is shown that if $j \geq 1$, then

$$|\hat{\lambda}_j - \lambda_j| \leq \hat{\Delta}_{\mathbf{Z}}, \tag{32}$$

and that if $j \leq \hat{R}_N^{\mathbf{Z}}$ (recall the definition of $\hat{R}_N^{\mathbf{Z}}$ in (17)), then

$$\|\hat{\phi}_j - \phi_j\| \leq 8^{1/2} \hat{\Delta}_{\mathbf{Z}} (\delta_j^{\mathbf{Z}})^{-1}, \tag{33}$$

$$\hat{\Delta}_{\mathbf{Z}} = O_p(N^{-1/2}), \tag{34}$$

$$R_N^{\mathbf{Z}} \to \infty \text{and } \hat{R}_N^{\mathbf{Z}} \leq \hat{\lambda}_1^{\mathbf{Z}} \eta_N^{-1}. \tag{35}$$

Moreover, if $j \leq \hat{R}_N^{\mathbf{Z}}$, there exists a $k \leq j$ such that

$$\delta_j^{\mathbf{Z}} = \lambda_k - \lambda_{k+1} \geq \hat{\lambda}_k - \hat{\lambda}_{k+1} - 2\hat{\Delta}_{\mathbf{Z}} \geq \eta_N - 2\hat{\Delta}_{\mathbf{Z}} = \eta_N + o_P(\eta_N), \tag{36}$$

where we have applied (32) and (17). Using (34) and the assumption on $\eta_N$, we can conclude that $\eta_N > 2\hat{\Delta}_{\mathbf{Z}}$ from an index onward. Thus, (36) and (33) yield

$$\|\hat{\phi}_j - \phi_j\| \leq 8^{1/2} \frac{\hat{\Delta}_{\mathbf{Z}}}{\eta_N - 2\hat{\Delta}_{\mathbf{Z}}}. \tag{37}$$

From (32), (17) and (34), we obtain that

$$\lambda_j \geq \hat{\lambda}_j - \hat{\Delta}_{\mathbf{Z}} \geq \eta_N - \hat{\Delta}_{\mathbf{Z}} = \eta_N + o_P(\eta_N). \tag{38}$$

Now, we are in a position to prove Theorem 3.1.

*Proof of Theorem 3.1:* The proof is based on Lemma I.1. The result follows trivially from this lemma, the fact that $\hat{R}_N^{\mathbf{Z}} \xrightarrow{P} \infty$ as $n \to \infty$, and the result in Theorem 2.2.

**Lemma I.1** *Under the assumptions in Theorem 3.1, it happens that*

$$\left| \hat{D}_{\hat{R}_N^{\mathbf{Z}}}(\mathbf{Z}_1, \mathbf{Z}_2) - D_{\hat{R}_N^{\mathbf{Z}}}(\mathbf{Z}_1, \mathbf{Z}_2) \right| \xrightarrow{P} 0 \text{ as } n \to \infty.$$

*Proof.* For fixed $\mathbf{Z}_1, \mathbf{Z}_2$, let us denote $\mathbf{u} = \mathbf{Z}_1 - \mathbf{Z}_2$. Obviously, $\|\mathbf{u}\| = O(1)$ a.s. Let us denote $(u_1, \ldots, u_{\hat{R}_n^{\mathbf{Z}}})^T$ and $(\hat{u}_1, \ldots, \hat{u}_{\hat{R}_n^{\mathbf{Z}}})^T$ to be the projections of $\mathbf{u}$ on the subspaces generated by the first $\hat{R}_N^{\mathbf{Z}}$ eigenvectors of the matrices $\Sigma_{\hat{R}_N^{\mathbf{Z}}}^{\mathbf{Z}}$ and $\hat{\Sigma}_{\hat{R}_N^{\mathbf{Z}}}^{\mathbf{Z}}$, respectively, when written in the basis generated by those eigenvectors. Now, define

$$\hat{u}_j^{\mathbf{Z}} = \langle \mathbf{u}, \hat{\phi}_j^{\mathbf{Z}} \rangle = \int_0^1 \mathbf{u}(t) \hat{\phi}_j^{\mathbf{Z}}(t) dt$$

and similarly define $u_j$ for $j \in \mathbb{N}$. Fix $n \in \mathbb{N}$ and take $j \leq \hat{R}_N^{\mathbf{Z}}$. We now have

$$\left| \frac{(u_j)^2}{\lambda_j} - \frac{(\hat{u}_j)^2}{\hat{\lambda}_j} \right| = \left| \frac{u_j}{(\lambda_j)^{1/2}} - \frac{\hat{u}_j}{(\hat{\lambda}_j)^{1/2}} \right| \left| \frac{u_j}{(\lambda_j)^{1/2}} + \frac{\hat{u}_j}{(\hat{\lambda}_j)^{1/2}} \right|$$

$$\leq \left( \left| \frac{u_j - \hat{u}_j}{(\lambda_j)^{1/2}} \right| + \left| \hat{u}_j \frac{(\lambda_j)^{1/2} - (\hat{\lambda}_j)^{1/2}}{(\lambda_j \hat{\lambda}_j)^{1/2}} \right| \right) \left| \frac{u_j}{(\lambda_j)^{1/2}} + \frac{\hat{u}_j}{(\hat{\lambda}_j)^{1/2}} \right|.$$

We analyze each term in this expression separately as follows:

$$\left| \frac{u_j - \hat{u}_j}{(\lambda_j)^{1/2}} \right| \leq \frac{1}{(\lambda_j)^{1/2}} \int_0^1 |\mathbf{u}(t)| |\phi_j(t) - \hat{\phi}_j(t)| dt$$

$$\leq \frac{\|\mathbf{u}\| \, \|\phi_j - \hat{\phi}_j\|}{(\lambda_j)^{1/2}}$$

$$\leq 8^{1/2} \|\mathbf{u}\| \frac{\hat{\Delta}_{\mathbf{Z}}}{(\lambda_j)^{1/2}(\eta_n - 2\hat{\Delta}_{\mathbf{Z}})}$$

$$\leq 8^{1/2} \|\mathbf{u}\| \hat{\Delta}_{\mathbf{Z}} (\eta_n^{-3/2} + o_P(\eta_n^{-3/2})), \tag{39}$$

where we have applied the Cauchy-Schwarz inequality, (37), (34) and (38). On the other hand, we have

$$
\begin{aligned}
\left| \hat{u}_j \frac{(\lambda_j)^{1/2} - (\hat{\lambda}_j)^{1/2}}{(\lambda_j \hat{\lambda}_j)^{1/2}} \right| &\leq \int_0^1 |\mathbf{u}(t)||\hat{\phi}_j(t)| dt \frac{|\lambda_j - \hat{\lambda}_j|}{\left( (\lambda_j)^{1/2} + (\hat{\lambda}_j)^{1/2} \right)(\lambda_j \hat{\lambda}_j)^{1/2}} \\
&\leq \|\mathbf{u}\| \frac{\hat{\Delta}_{\mathbf{Z}}}{\left( (\lambda_j)^{1/2} + (\hat{\lambda}_j)^{1/2} \right)(\lambda_j \hat{\lambda}_j)^{1/2}} \\
&\leq \frac{1}{2} \|\mathbf{u}\| \hat{\Delta}_{\mathbf{Z}} (\eta_n^{-3/2} + o_P(\eta_n^{-3/2})),
\end{aligned}
\tag{40}
$$

where we have applied (17) and (38). Concerning the final term, using (38) and (17) again, we obtain that

$$
\left| \frac{u_j}{(\lambda_j)^{1/2}} + \frac{\hat{u}_j}{(\hat{\lambda}_j)^{1/2}} \right| \leq \|\mathbf{u}\| \left( \frac{1}{(\lambda_j)^{1/2}} + \frac{1}{(\hat{\lambda}_j)^{1/2}} \right) \leq \|\mathbf{u}\|(\eta_n^{-1/2} + o_P(\eta_n^{-1/2})).
\tag{41}
$$

Now, define $C = 8^{1/2} + 1$. Combining (39), (40), (41), (35) and (34), we get the following:

$$
\begin{aligned}
\left| \hat{D}_{\hat{R}_N^{\mathbf{Z}}}(\mathbf{Z}, \mathbf{Z}_2) - D_{\hat{R}_N^{\mathbf{Z}}}(\mathbf{Z}, \mathbf{Z}_2) \right| &\leq \frac{1}{\hat{R}_N^{\mathbf{Z}}} \sum_{j=1}^{\hat{R}_N^{\mathbf{Z}}} \left| \frac{(u_j)^2}{\lambda_j} - \frac{(\hat{u}_j)^2}{\hat{\lambda}_j} \right| \\
&\leq C\|\mathbf{u}\|^2 \hat{\Delta}_{\mathbf{Z}}(\eta_n^{-2} + o_P(\eta_n^{-2})) = O_P(n^{-1/2}\eta_n^{-2}).
\end{aligned}
$$

By construction, $\eta_n$ is such that $n\eta_n^5 \to \infty$. So, we have $|\hat{D}_{\hat{R}_N^{\mathbf{Z}}}(\mathbf{Z}_1, \mathbf{Z}_2) - D_{\hat{R}_N^{\mathbf{Z}}}(\mathbf{Z}_1, \mathbf{Z}_2)| \xrightarrow{P} 0$ as $n \to \infty$, and the lemma is proved. ●

## J  Proof of Theorem 3.2

We now prove the following lemma.

**Lemma J.1** *Under the assumptions in Theorem 3.2, we have that* $\mathbb{P}[\hat{R}_N \geq R_N] \to 1$ *as* $N \to \infty$.

*Proof*: Fix $N \in \mathbb{N}$. From (32), we have that

$$
\inf_{j \leq R_N} (\hat{\lambda}_j - \hat{\lambda}_{j+1}) \geq \inf_{j \leq R_N} (\lambda_j - \lambda_{j+1}) - 2\hat{\Delta}_{\mathbf{Z}} \geq (1 + \delta)\eta_N - 2\hat{\Delta}_{\mathbf{Z}},
$$

and the proof ends because (34) and the fact that $\eta_N \geq N^{-1/5}$ imply that $\mathbb{P}[\delta\eta_N - 2\hat{\Delta}_{\mathbf{Z}} \geq 0] \to 1$. ●

In this setting, recall that $L_\mu = 0$ and $L_S < \infty$ (see part (b) of Theorem 2.2). We will only prove part b); part a) being similar. W.l.o.g. we will assume that $h = 1$ and $k = 2$. Recall that for every $\mathbf{Z}_1$ and $\mathbf{Z}_2$, we have

$$
D_{\hat{R}_N}(\mathbf{Z}_1, \mathbf{Z}_2) = \frac{1}{\hat{R}_N} \sum_{j=1}^{\hat{R}_N} \frac{\langle \mathbf{Z}_1 - \mathbf{Z}_2, \phi_j \rangle^2}{\lambda_j} \text{ and } \hat{D}_{\hat{R}_N}(\mathbf{Z}_1, \mathbf{Z}_2) = \frac{1}{\hat{R}_N} \sum_{j=1}^{\hat{R}_N} \frac{\langle \mathbf{Z}_1 - \mathbf{Z}_2, \hat{\phi}_j \rangle^2}{\hat{\lambda}_j}.
$$

We are going to consider the function

$$
\tilde{D}_{\hat{R}_N}(\mathbf{Z}_1, \mathbf{Z}_2) = \frac{1}{\hat{R}_N} \sum_{j=1}^{\hat{R}_N} \frac{\langle \mathbf{Z}_1 - \mathbf{Z}_2, \phi_j \rangle^2}{\hat{\lambda}_j}.
$$

Obviously,

$$
\begin{aligned}
\sup_{\mathbf{Z}_1 \in \mathcal{C}_1^N, \mathbf{Z}_2 \in \mathcal{C}_2^N} \left| \hat{D}_{\hat{R}_N}(\mathbf{Z}_1, \mathbf{Z}_2) - L_S^{12} \right| \quad \leq \quad & \sup_{\mathbf{Z}_1 \in \mathcal{C}_1^N, \mathbf{Z}_2 \in \mathcal{C}_2^N} \left| \hat{D}_{\hat{R}_N}(\mathbf{Z}_1, \mathbf{Z}_2) - \tilde{D}_{\hat{R}_N}(\mathbf{Z}_1, \mathbf{Z}_2) \right| \\
& + \sup_{\mathbf{Z}_1 \in \mathcal{C}_1^N, \mathbf{Z}_2 \in \mathcal{C}_2^N} \left| \tilde{D}_{\hat{R}_N}(\mathbf{Z}_1, \mathbf{Z}_2) - D_{\hat{R}_N}(\mathbf{Z}_1, \mathbf{Z}_2) \right| \\
& + \sup_{\mathbf{Z}_1 \in \mathcal{C}_1^N, \mathbf{Z}_2 \in \mathcal{C}_2^N} \left| D_{\hat{R}_N}(\mathbf{Z}_1, \mathbf{Z}_2) - L_S^{12} \right| \\
=: \quad & T_1 + T_2 + T_3.
\end{aligned}
$$

Lemma J.1, and equations (35) and (32) imply that there exists $C > 0$ such that

$$
\mathbb{P}[R_N \leq \hat{R}_N \leq C N^{1/5}] \to 1.
$$

Consequently, with probability going to 1, it happens that

$$
D_{R_N}(\mathbf{Z}_1, \mathbf{Z}_2) \leq D_{\hat{R}_N}(\mathbf{Z}_1, \mathbf{Z}_2) \leq D_{C N^{1/5}}(\mathbf{Z}_1, \mathbf{Z}_2).
$$

Since, by assumption (15), $\log N = o\left(\frac{R_N}{\lambda_1}\right)$ and trivially we have $\log N = o\left(\frac{C N^{1/5}}{\lambda_1}\right)$, b) in Theorem 2.6 gives that $T_3$ converges in probability to zero as $N \to \infty$. Since $L_S^{12} < \infty$, this fact implies that

$$
\sup_{\mathbf{Z}_1 \in \mathcal{C}_1^N, \mathbf{Z}_2 \in \mathcal{C}_2^N} D_{\hat{R}_N}(\mathbf{Z}_1, \mathbf{Z}_2) = O_P(1). \tag{42}
$$

With respect to $T_2$, we have

$$
\begin{aligned}
T_2 \quad \leq \quad & \sup_{\mathbf{Z}_1 \in \mathcal{C}_1^N, \mathbf{Z}_2 \in \mathcal{C}_2^N} \frac{1}{\hat{R}_N} \sum_{j=1}^{\hat{R}_N} \frac{\langle \mathbf{Z}_1 - \mathbf{Z}_2, \phi_j \rangle^2}{\lambda_j} \frac{|\lambda_j - \hat{\lambda}_j|}{\hat{\lambda}_j} \\
\leq \quad & \sum_{j=1}^{\hat{R}_N} \frac{|\lambda_j - \hat{\lambda}_j|}{\hat{\lambda}_j} \sup_{\mathbf{Z}_1 \in \mathcal{C}_1^N, \mathbf{Z}_2 \in \mathcal{C}_2^N} D_{\hat{R}_N}(\mathbf{Z}_1, \mathbf{Z}_2) = O_p(N^{-1/10}),
\end{aligned}
$$

where last equality follows from (42), (32), (34), (35) and (17).

Finally, given $\mathbf{Z}_1 \in \mathcal{C}_1^N, \mathbf{Z}_2 \in \mathcal{C}_2^N$, the Cauchy-Schwarz inequality and the fact that $\|\hat{\phi}_j\| = \|\phi_j\| = 1$ imply

$$
\begin{aligned}
\left| \hat{D}_{\hat{R}_n}(\mathbf{Z}_1, \mathbf{Z}_2) - \tilde{D}_{\hat{R}_n}(\mathbf{Z}_1, \mathbf{Z}_2) \right| \quad \leq \quad & \frac{1}{\hat{R}_N} \sum_{j=1}^{\hat{R}_N} \frac{\left| \langle \mathbf{Z}_1 - \mathbf{Z}_2, \hat{\phi}_j \rangle^2 - \langle \mathbf{Z}_1 - \mathbf{Z}_2, \phi_j \rangle^2 \right|}{\hat{\lambda}_j} \\
= \quad & \frac{1}{\hat{R}_N} \sum_{j=1}^{\hat{R}_N} \frac{\left| \langle \mathbf{Z}_1 - \mathbf{Z}_2, \hat{\phi}_j - \phi_j \rangle \right| \left| \langle \mathbf{Z}_1 - \mathbf{Z}_2, \hat{\phi}_j + \phi_j \rangle \right|}{\hat{\lambda}_j} \\
\leq \quad & \|\mathbf{Z}_1 - \mathbf{Z}_2\|^2 \frac{1}{\hat{R}_N} \sum_{j=1}^{\hat{R}_N} \frac{\|\hat{\phi}_j - \phi_j\| \, \|\hat{\phi}_j + \phi_j\|}{\hat{\lambda}_j} \\
\leq \quad & 2\|\mathbf{Z}_1 - \mathbf{Z}_2\|^2 \frac{1}{\hat{R}_N} \sum_{j=1}^{\hat{R}_N} \frac{\|\hat{\phi}_j - \phi_j\|}{\hat{\lambda}_j} \\
= \quad & 2\|\mathbf{Z}_1 - \mathbf{Z}_2\|^2 H_N.
\end{aligned}
$$

Moreover, the application of (33), (34), (36) and (17) gives that $H_N = O_P(N^{-1/10})$, which in turn is equivalent to saying that there exists $C > 0$ such that $\mathbb{P}[H_n < C N^{-1/10}] \to 1$. This and the reasoning leading to (28) imply that to prove $T_1 \xrightarrow{P} 0$, it is enough to show that for every $C > 0$

$$
N^2 \mathbb{P}\left[ \|\mathbf{Z}_1 - \mathbf{Z}_2\|^2 > C N^{1/10} \right] \to 0 \text{ as } N \to \infty, \tag{43}
$$

where $\mathbf{Z}_1$ and $\mathbf{Z}_2$ came from distributions $\mathbb{P}_1$ and $\mathbb{P}_2$, respectively.

To show (43), notice that $\mathbf{Z}_1 - \mathbf{Z}_2$ follows a Gaussian distribution whose mean function is $\mu^{\mathbf{Z}_1} - \mu^{\mathbf{Z}_2}$ and its covariance is $\Sigma^{12} = \Sigma^{\mathbf{Z}_1} + \Sigma^{\mathbf{Z}_2}$. Let $\gamma_j$ with $j \in \mathbb{N}$ denote the ordered eigenvalues of $\Sigma^{12}$. Consider a basis composed by the eigenfunctions of $\Sigma^{12}$, we denote by $(\mu^{\mathbf{Z}_1} - \mu^{\mathbf{Z}_2})_j$ the components of $\mu^{\mathbf{Z}_1} - \mu^{\mathbf{Z}_2}$ in this basis and $\{u_j\}_{j \in \mathbb{N}}$ is a sequence of i.i.d. real standard normal variables. Now, we have the following

$$
\begin{aligned}
\|\mathbf{Z}_1 - \mathbf{Z}_2\|^2 \quad &\sim \quad \sum_{j=1}^{\infty} \left( \gamma_j^{1/2} u_j + (\mu^{\mathbf{Z}_1} - \mu^{\mathbf{Z}_2})_j \right)^2 \\
&= \quad \sum_{j=1}^{\infty} \left( \gamma_j(u_j^2 - 1) + \gamma_j + (\mu^{\mathbf{Z}_1} - \mu^{\mathbf{Z}_2})_j^2 + 2(\mu^{\mathbf{Z}_1} - \mu^{\mathbf{Z}_2})_j \gamma_j^{1/2} u_j \right) \\
&= \quad \sum_{j=1}^{\infty} \left( \gamma_j(u_j^2 - 1) + 2(\mu^{\mathbf{Z}_1} - \mu^{\mathbf{Z}_2})_j \gamma_j^{1/2} u_j \right) + \text{trace}(\Sigma^{12}) + \|\mu^{\mathbf{Z}_1} - \mu^{\mathbf{Z}_2}\|^2.
\end{aligned}
$$

Note that $K := \text{trace}(\Sigma^{12}) + \|\mu^{\mathbf{Z}_1} - \mu^{\mathbf{Z}_2}\|^2 < \infty$. Thus,

$$
\begin{aligned}
\mathbb{P}\left[ \|\mathbf{Z}_1 - \mathbf{Z}_2\|^2 > CN^{1/10} \right] &= \quad \mathbb{P}\left[ \sum_{j=1}^{\infty} \left( \gamma_j(u_j^2 - 1) + 2(\mu^{\mathbf{Z}_1} - \mu^{\mathbf{Z}_2})_j \gamma_j^{1/2} u_j \right) > CN^{1/10} - K \right] \\
&\leq \quad \mathbb{P}\left[ \sum_{j=1}^{\infty} \gamma_j(u_j^2 - 1) > \frac{1}{2}\left( CN^{1/10} - K \right) \right] \\
&\quad + \mathbb{P}\left[ \sum_{j=1}^{\infty}(\mu^{\mathbf{Z}_1} - \mu^{\mathbf{Z}_2})_j \gamma_j^{1/2} u_j > \frac{1}{4}\left( CN^{1/10} - K \right) \right] \\
&=: \quad P_1 + P_2.
\end{aligned}
\tag{44}
$$

Obviously, $\frac{1}{4}\left( CN^{1/10} - K \right) \to \infty$. Thus, eventually $\frac{1}{4}\left( CN^{1/10} - K \right) > 1$ and from Lemma G.1, we have that

$$
P_1 \leq \lim_{d \to \infty} \mathbb{P}\left[ \sum_{j=1}^{d} \gamma_j(u_j^2 - 1) > \frac{1}{2}\left( CN^{1/10} - K \right) \right] \leq 2\exp\left( -\frac{1}{8\gamma_1}\left( CN^{1/10} - K \right) \right).
\tag{45}
$$

For $P_2$, first note that the real r.v. $\sum_{j=1}^{d}(\mu^{\mathbf{Z}_1} - \mu^{\mathbf{Z}_2})_j \gamma_j^{1/2} u_j$ is centered normal, with variance equal to $\sum_{j=1}^{d}(\mu^{\mathbf{Z}_1} - \mu^{\mathbf{Z}_2})_j^2 \gamma_j \leq \gamma_1 \sum_{j=1}^{d}(\mu^{\mathbf{Z}_1} - \mu^{\mathbf{Z}_2})_j^2 \leq \gamma_1 \|\mu^{\mathbf{Z}_1} - \mu^{\mathbf{Z}_2}\|^2$ for every $d \in \mathbb{N}$. Therefore,

$$
\begin{aligned}
P_2 &\leq \quad \lim_{d \to \infty} \mathbb{P}\left[ \left| \sum_{j=1}^{d}(\mu^{\mathbf{Z}_1} - \mu^{\mathbf{Z}_2})_j \gamma_j^{1/2} u_j \right| > \frac{1}{4}\left( CN^{1/10} - K \right) \right] \\
&\leq \quad \mathbb{P}\left[ |N(0,1)| > \frac{1}{4\gamma_1^{1/2}\|\mu^{\mathbf{Z}_1} - \mu^{\mathbf{Z}_2}\|}\left( CN^{1/10} - K \right) \right] \\
&\leq \quad \sqrt{\frac{2}{\pi}} \exp\left( -\frac{1}{2\gamma_1(4\|\mu^{\mathbf{Z}_1} - \mu^{\mathbf{Z}_2}\|)^2}\left( CN^{1/10} - K \right)^2 \right),
\end{aligned}
\tag{46}
$$

where last inequality comes from (25) because, eventually $1 < \left( CN^{1/10} - K \right)/(4\gamma_1^{1/2}\|\mu^{\mathbf{Z}_1} - \mu^{\mathbf{Z}_2}\|)$. Finally, (44), (45), and (46) give (43) and consequently, $T_1 \xrightarrow{P} 0$ as $N \to \infty$. $\qquad\bullet$

## Appendix II: Additional Material

## K   Extension to non-Gaussian Processes

Obviously, non-Gaussian processes can also be mutually singular. In fact, Theorem 4.3 in Rao and Varadara-jan (1963) contains a sufficient condition for this property to be satisfied. This allows us to consider the possibility to extend previous results to cover non-Gaussian distributions. It is obvious that the proofs that we have developed can cover non-Gaussian distributions as long as they satisfy the due properties. In this subsection, we state the properties a distribution should satisfy in order the proofs can be extended. Let $\mathbb{P}_1$ and $\mathbb{P}_2$ be two probabilities on the Hilbert space $\mathbb{H}$. Here, $\mathbf{Z}$ will denote a $L_2[0,1]$-valued random element with distribution $\pi_1\mathbb{P}_1 + \pi_2\mathbb{P}_2$ for some $\pi_1, \pi_2 > 0$ with $\pi_1 + \pi_2 = 1$.

The basic assumption is the existence of a covariance of $\mathbf{Z}$. We will also consider assumptions *A.1* and *A.2* (see Section 3 of the paper) and $b \in L_2[0,1]$. Given a p.d. $d \times d$ matrix $A_d$ and a $d$-dimensional subspace $V_d \subset L_2[0,1]$, we need to consider the $d$-dimensional random vector $\mathbf{U}_d = (A_d)^{-1/2}(\mathbf{Z} - b)_d$ and the covariance matrix $S_d = A_d^{-1/2}\Sigma_d^{\mathbf{Z}}A_d^{-1/2}$, where $\Sigma_d^{\mathbf{Z}}$ is the covariance matrix of $\mathbf{Z}_d$ and $(\mathbf{Z} - b)_d$ is the projection on $V_d$ of $(\mathbf{Z} - b)$ with $d \in \mathbb{N}$.

Let us write $\mathbf{U}_d - \mathbb{E}[\mathbf{U}_d] = (u^1, \ldots, u^d)^T$ in the basis of the eigenvectors of $S_d$ and let $\alpha_1^d, \ldots, \alpha_d^d$ denote the eigenvalues of $S_d$. Therefore, $u_i/\alpha_i^d$ for $1 \le i \le d$ are real, standardised random variables which we need to assume to be i.i.d. Similar properties must hold for the decomposition of $\mathbf{Z}$ in its eigenfunction basis (also see Dai et al (2017)). We finally need two exponential inequalities as those stated in (25) of Lemma G.1.

## L   Discussion on GP Clustering for the 'Location Only' Case

We have some ideas to fix the problem with the 'location only' case. Recall the notation used in Subsection 2.1 of the paper. As stated there, the problem in this case is that

$$D_d^{\Sigma_d^{\mathbf{Z}}}(\mathbf{u}, \mathbf{v}) = \frac{1}{d}\left\|(\Sigma_d^{\mathbf{Z}})^{-1/2}(\mathbf{u} - \mathbf{v})_d\right\|^2 = \frac{1}{d}\sum_{i=1}^{d}\frac{(u_i - v_i)^2}{\lambda_i} \xrightarrow{P} 0 \text{ as } d \to \infty.$$

Our idea is to replace the terms in the sum with some others going to 0 slowly enough (or, if possible, not converging to zero at all). To use this idea, our proposal is as follows:

$$D_d^{\Sigma_d^{\mathbf{Z}},r}(\mathbf{u}, \mathbf{v}) := \frac{1}{d}\left\|((\Sigma_d^{\mathbf{Z}})^{-1/2})^r(\mathbf{u} - \mathbf{v})_d\right\|^2 = \frac{1}{d}\sum_{i=1}^{d}\frac{(u_i - v_i)^2}{\lambda_i^r}, \text{ with } r \in \mathbb{I}. \tag{47}$$

Here, $\mathbb{I}$ is the set of integers. In this article, we have studied the case when $r = 1$, i.e., $D_d^{\Sigma_d^{\mathbf{Z}},1}$. However, this was not a strict requirement and we look into some possible scenarios below:

- If $r \in \{0, -1, -2, \ldots\}$, then assumption *A.2* in the main paper trivially gives that $\frac{(u_i - v_i)^2}{\lambda_i^r} \le \frac{(u_i - v_i)^2}{\lambda_i}$ eventually for large $i$. Consequently, we have $D_d^{\Sigma_d^{\mathbf{Z}},r}(\mathbf{u}, \mathbf{v}) \xrightarrow{P} 0$ as $d \to \infty$.

- When $r \in \{2, 3, \ldots\}$, the transformation $D_d^{\Sigma_d^{\mathbf{Z}},r}$ may be useful because $1/\lambda_i^r$ will start to take high values (recall assumption *A.2*) and this may lead to separation between the observations of corresponding to different clusters.

Keeping the viewpoint stated above in mind, we consider the transformation $D_d^{\Sigma_d^{\mathbf{Z}},4}$ (using $r = 4$ in (47)). Numerical results for the difference in `location only` setting stated in Section 4 are reported below. We have excluded Example II from this comparison because, as stated earlier, the two GPs have no differences in their means.

Table L.1: Adjusted Rand distances for different GPs with differences only in locations (with standard error in brackets).

| Ex. | $k$-means | spectral | mclust | funclust | CL | DHP | CD | | |
|-----|-----------|----------|--------|----------|----|-----|-----------|----------|--------|
| | | | | | | | $k$-means | spectral | mclust |
| I | **0.0001** | **0.0001** | 1.0000 | 0.0002 | **0.0001** | **0.0001** | *0.0012* | 0.0814 | 0.0016 |
| | (0.0001) | (0.0001) | (0.0000) | (0.0001) | (0.0000) | (0.0001) | (0.0002) | (0.0072) | (0.0003) |
| III | **0.0646** | 0.0960 | 1.0000 | *0.0795* | 0.0945 | 0.1480 | 0.1965 | 0.1948 | 0.1649 |
| | (0.0015) | (0.0018) | (0.0000) | (0.0009) | (0.0045) | (0.0047) | (0.0025) | (0.0027) | (0.0017) |
| IV | 0.1606 | 0.2111 | 1.0000 | *0.0318* | 0.1015 | **0.0134** | 0.1257 | 0.3777 | 0.1784 |
| | (0.0007) | (0.0008) | (0.0000) | (0.0003) | (0.0000) | (0.0004) | (0.0019) | (0.0085) | (0.0021) |

As expected, the performance of $k$-means is quite good in Examples I and III. Both DHP and CL also perform quite well securing a first place in some cases. The proposed statistic $D_d^{\Sigma_d^{\mathbf{Z}},4}$ shows significant improvement (recall from part (b) of Theorem 2.2 that $L_\mu = 0$ for the earlier transformation $D_d^{\Sigma_d^{\mathbf{Z}},1}$), and this is reflected in the numerical figures of Table L.1.

Clearly, there is scope of further work with the proposed transformation $D_d^{\Sigma_d^{\mathbf{Z}},r}$ for $r \in \{2, 3, \ldots\}$, both theoretically as well as numerically.

## M   Review of the paper by Delaigle et al (2019)

As stated, Delaigle et al (2019) is related to *perfect clustering* and it is the only paper on perfect clustering we are aware of. In this section, we analyze the relation between our proposal and this paper.

The proposal by Delaigle et al (2019) is based on finding a finite-dimensional subspace in which the data are projected, and clustering is done by applying a modification of the $k$-means algorithm on those projections. A theoretical result related to perfect clustering is stated in Theorem 1 of this paper. In the homoscedastic case, Delaigle et al (2019) gives an explicit expression of the subspace in which the data should be projected (see Theorem 2 of this paper).

The technique proposed in this paper has some advantages over our proposal in the sense that they can `handle the homoscedastic (differences only in location) case`. However, it suffers from several limitations, the main being `its difficulty to deal with more than two populations`. As stated in Section 5.2 of Delaigle et al (2019), *"...it is not clear how to extend...* [their method] *...to more than two populations."*. The only exception being the case in which one needs to assume that the data follow a *binary hierarchical structure*, where the groups can be sequentially split into two. As a consequence of this, Delaigle et al (2019) `do not propose a procedure to estimate the true number of clusters`.

`On the technical side, the theory of` Delaigle et al (2019) `has some limitations`. It requires to arbitrarily fix $p \in \mathbb{N}$. Then, the data are projected on a $p$-dimensional subspace in which the clustering is to be done. New issues appear in the way in which the subspace should be chosen as well as the clusters need to be constructed. According to Theorem 1 of this paper, the generators of the subspace must be chosen in a finite set with cardinality $a_n \to \infty$ as the sample size $n \to \infty$. Moreover, the partition of the data set must be chosen between those in a finite set of Voronoi tessellations of $\mathbb{R}^p$ with cardinality $b_n \to \infty$ as $n \to \infty$. Additionally, the result needs technical conditions like the existence of some $c \in (0, 1)$ such that for every $C > 0$ it happens that $a_n^p b_n \exp(-Cn^c) \to \infty$ as $n \to \infty$.

## N    Full Numerical Results

Full results for `two scenarios` are given below.

Table N.1: Adjusted Rand distances for different GPs with differences in locations and scales (with standard error in brackets).

| Ex. | CL1 | CL2 | DHP1 | DHP2 |
|-----|-----|-----|------|------|
| I | 0.0239 | 0.0554 | 0.8386 | 0.0818 |
| | (0.0007) | (0.0005) | (0.0064) | (0.0025) |
| II | 0.5767 | 0.9967 | 0.5470 | 0.5149 |
| | (0.0045) | (0.0018) | (0.0047) | (0.0049) |
| III | 0.2891 | 0.9962 | 0.4137 | 0.5613 |
| | (0.0000) | (0.0000) | (0.0054) | (0.0060) |
| IV | 0.1833 | 0.6660 | 0.1379 | 0.5211 |
| | (0.0000) | (0.0000) | (0.0033) | (0.0061) |

Table N.2: Adjusted Rand distances for different GPs with differences only in scales (with standard error in brackets).

| Ex. | CL1 | CL2 | DHP1 | DHP2 |
|-----|-----|-----|------|------|
| I | 0.8269 | 1.0017 | 0.9989 | 0.9966 |
| | (0.0000) | (0.0000) | (0.0005) | (0.0008) |
| II | 0.9065 | 1.0019 | 1.0001 | 0.9999 |
| | (0.0007) | (0.0005) | (0.0002) | (0.0003) |
| III | 0.9994 | 0.9998 | 0.9984 | 0.9967 |
| | (0.0000) | (0.0000) | (0.0004) | (0.0007) |
| IV | 0.8464 | 0.9928 | 0.9994 | 0.9980 |
| | (0.0000) | (0.0000) | (0.0003) | (0.0006) |

Full result for the `location only` case (using the transformation $D_d^{\Sigma_d^{\mathbf{Z}},4}$ stated in Section L of Appendix II) is given next.

Table N.3: Adjusted Rand distances for different GPs with differences in locations (with standard error in brackets).

| Ex. | C1 | C2 | DHP1 | DHP2 |
|-----|-----|-----|------|------|
| I | 0.0001 | 0.0001 | 0.9896 | 0.0001 |
| | (0.0000) | (0.0000) | (0.0001) | (0.0001) |
| III | 0.0945 | 0.9975 | 0.1480 | 0.1623 |
| | (0.0010) | (0.0043) | (0.0047) | (0.0075) |
| IV | 0.1015 | 0.9001 | 0.0134 | 0.1473 |
| | (0.0000) | (0.0000) | (0.0004) | (0.0039) |

R codes for our clustering methods are available from this link: GP clustering.

## O   Pseudo-codes of Our Methods

---

**Algorithm 1** Clustering Algorithm

---
Data: $Z_1, \ldots, Z_N$ are observed at $p$ time points (i.e., $p$-dimensional data);
Compute the sample variance-covariance matrix $S_N$;                                                   $\triangleright\ O(Np^2)$
Define $d_N = \min\{p, N\}$;
Spectral decomposition of the $d_N \times d_N$ matrix $S_N$ yields $e_1, \ldots, e_{d_N}$ (eigen functions) and $\lambda_1, \ldots, \lambda_{d_N}$
(eigen values);                                                                                         $\triangleright\ O(p^3)$
Fix $1 \le d \le d_N$;
**for** $i = 1 : N$ **do**
    **for** $j = 1 : d$ **do**
        Compute $z_{ij} = \langle Z_i, e_j \rangle$;
    **end for**
**end for**                                                                                             $\triangleright\ O(N^2 p)$
**for** $i = 1 : N$ **do**
    **for** $i' = 1 : N$ **do**
        Compute $D_d(Z_i, Z_i') = \dfrac{1}{d} \sum_{k=1}^{d} \dfrac{(z_{ij} - z_{i'j})^2}{\lambda_k}$;
    **end for**
**end for**                                                                                             $\triangleright\ O(N^2 d)$
**for** $i = 1 : N$ **do**
    **for** $i' = 1 : N$ **do**
        $temp = 0$;
        **for** $t \ne i, i'$ **do**
            Compute $temp = temp + [D_d(Z_i, Z_t) - D_d(Z_i', Z_t)]^2$;
        **end for**
        $\gamma_d(Z_i, Z_i') = temp$;
    **end for**
**end for**                                                                                             $\triangleright\ O(N^2 d)$
Spectral decomposition of the $N \times N$ matrix $\Gamma_N$ yields $\beta_1, \ldots, \beta_N$ (eigen values);     $\triangleright\ O(N^3)$

Use `optishrink` to get $K_d$;
Implement any suitable clustering algorithm on $\Gamma_N$ with $K_d$ as input for the number of clusters;
Obtain class labels for the $N$ observations, or a given data set.

---

**Algorithm 2** Cross Validation Algorithm to Choose the Value of $d$

---
Consider the data $S = \{Z_1, \ldots, Z_N\}$;
**for** $d = 2 : d_N$ **do**;
    Fix $B \in \mathbb{N}$;
    **for** $b = 1 : B$ **do**
        Split the set $S$ into three disjoint subsets $S_{1b}$, $S_{2b}$ and $S_{3b}$;
        Use Algorithm 1 with data points corresponding to $S_{1b}$ to get a cluster assignment for data points
corresponding to $S_{3b}$ (say, $C_1$);
        Use Algorithm 1 with data points corresponding to $S_{2b}$ to get a cluster assignment for data points
corresponding to $S_{3b}$ (say, $C_2$);
        Compute a distance between $C_1$ and $C_2$ (say, $\mathbb{D}_b$)
    **end for**                                                                                 $\triangleright\ O(Bd_N)$
    Average over $\mathbb{D}_1, \ldots, \mathbb{D}_B$ to get $\mathbb{D}_d^{CV}$;                         $\triangleright\ O(d_N)$
**end for**
Return $d_{CV} = \arg\min_{2 \le d \le d_N} \mathbb{D}_d^{CV}$.

---

