# OpenReview forum: "On Perfect Clustering for Gaussian Processes"
_TMLR — Accepted by TMLR_

### Review · Reviewer_BSFc · 2023-07-20

**Summary Of Contributions:**

This paper first studied some transformation for Gaussian processes and its limit theorem through the lens of mutually singularity and then showed that it is possible to achieve perfect separations of mixtures of Gaussian processes through running common clustering algorithms like k-means on an appropriate modification of this transformation especially for the two components cases. This method performs better when comparing with different baseline methods on simulated and benchmark datasets.

**Audience:**

Yes

**Broader Impact Concerns:**

None.

**Claims And Evidence:**

No

**Requested Changes:**

On the presentation side,
- Please consider rewriting the Abstract to not over-claim anything: the limitations of handling "difference only in location" case and if the proposed method only leads to perfect separation in 2-mixture. If the proposed method does lead to perfect separation in J-mixture with J>2, I would suggest having a dedicated section in the main text detailing the steps achieving that. The method used for 2-mixture does not seem to work with more than 2 components.
- Please consider adding the problem formulation studied in this work along with the definition of perfect clustering in the Introduction, and including a Contributions subsection to make it easier for the readers to follow.
- Please consider improving the presentation of technical part especially adding the full specification of the assumptions/conditions and providing more intuitive explanations for the Remarks. For example, it is not quite straightforward to see why b = 0 used in Remark 2.1.3 would satisfy all the conditions of Theorem 2.1.
- Please consider improving the discussion in Remark 2.5.2. The argument of "the structure ... will lead us to perfect clustering for every value of J $\ge$ 2" is not supported with any concrete method since the procedure described in the first paragraph on page 9 does not produce a perfect clustering. It also argues that "the procedure described in Proposition 2.3 also works fine, with the limit equal to the rank of $\Gamma$", but the following paragraph mentioned that the rank does not generally coincides with the number of clusters.
- Please consider adding the full description of the empirical implementations in Section 3.2 with all the details of estimating the dimension and the number of clusters. It would also be better to highlight the differences between the theoretical derivations and the empirical implementations.
- Please consider correcting the discussion of Delaigle et al (2019) since the work extends their theoretical results to the case of more than two populations.

On the evaluation side,
- Please consider including visualizations of typical curves from each group for each case to better illustrate the experimental settings and challenges.
- Please consider addressing the inconsistencies between theoretical derivations and empirical results discussed above.
- Please consider addressing the unfair baseline comparisons and insufficient validations discussed above.
- Please consider adding empirical validation on working with more than 2 clusters depending on if the authors want to claim the perfect clustering in this scenario.

**Strengths And Weaknesses:**

Strengths:
- This paper addressed an interesting problem: how to perfectly recover the component assignment for data generated from a mixture of Gaussian processes especially for the 2-component case through the lens of mutually singularity.

- The authors provided solid theoretical foundations for both the proposed transformation and its application to Gaussian processes clustering, which lead to both estimation of the unknown number of clusters and the perfect separation of populations.

- The experiments showcased the effectiveness of the proposed method comparing to different baseline methods.

Weaknesses:
- There is no proper definition of perfect clustering in the paper. As shown in [Priebe et. al. 2019], there is no universal definition of what a perfect clustering is. It seems that it refers to perfectly recover the component assignment if the data is generated from a mixture of Gaussian processes according to Definition 2.4 and Theorem 2.5. However, this definition cannot be directly applied to the J-mixture case with J > 2 since the proposed method produces more than J clusters as described in the first paragraph on page 9 which will not result an adjusted rand index approaching 1. Thus, it is unclear how to apply the proposed method to more than 2 clusters to achieve perfect clustering and the description presented is quite confusing. Also, there is no empirical validation on working with more than 2 clusters.

- It would be better to specify the full set of assumptions/conditions for the main theorems. For example, the conditions used in Theorem 2.2 have been referenced multiple times in the paper, but they are not fully specified in the theorem. Additional assumptions were discussed in Section 2.1 as well. On the other hand, it is unclear if the choice of the $V_d$ as "the d-dimensional subspaces generated by the d eigenfunctions associated with the d largest eigenvalues of $\Sigma^{Z}$" is critical to Theorem 2.2.

- There lacks detailed full description of the proposed clustering algorithm. Even though the description in Section 3.2 appears quite simple, the actual implementation is much more involved as described in Section 4.2. It would be much better to have a single detailed description of the full implementation including ways to estimate the dimension d and the number of clusters J and highlight the differences between the theoretical derivations and empirical implementations.

- Inconsistencies between theoretical derivations and empirical results. The derivations in Section 2 and 3 suggest that any clustering method can be applied on top of the transformed data to get perfect clustering and the authors also called out the k-means clustering there. However, there are large performance variations among different clustering methods applied to the transformed data especially in Example II and III as shown in Table N where k-means is usually the worst-performing one. This is particularly noteworthy since the theoretical analysis does not suggest such disparities. Unfortunately, the authors did not provide any discussion about why and how this happens in practice.

- Unfair baseline comparisons and insufficient validations. First, the quantitative results reported in Section 4 and 5 are the best among 3 different clustering methods, which essentially does model selection on the test data. A more apples-to-apples comparison would be to compare k-means with CD-k-means and GMM with CD-mcluster, since the proposed method is focused on the transformed data not the clustering method on top. But when comparing k-means results in Table 4.1 with CD-k-means in Table N.1, we can see that vanilla k-means is actually performing better or comparable in three settings. Furthermore, the authors only considered the case with fixed number of samples and equal mixing weights of each component. It would be nicer to include results with different numbers of samples and varied mixing weights to showcase the asymptotic properties especially the relationship between N and d and handling of unequal mixing weights. Lastly, the proposed method seems to perform better in the "difference in only scales" case than the "difference in location and scales" case, even though the latter is an easier one. The authors might provide some discussions regarding this situation.


Priebe et. al. 2019. On a two-truths phenomenon in spectral graph clustering. PNAS.

---

> ### Author Response · Authors · 2023-08-14
> **Revised version and response**
>
> Thank you for your constructive comments!
>
> We are now submitting a revised submission. All the changes in this revised version are now marked in Red color.
>
> A point by point response to the requested changes is in the file "RoR.pdf" (under Supplementary Material).

---

> > ### Comment · Reviewer_BSFc · 2023-08-14
> >
> > Thanks for the response and the revised manuscript, especially the improved presentation and the additional quantitative experiments with J > 2. These have addressed most of the concerns.

---

> > > ### Author Response · Authors · 2023-08-28
> > >
> > > Thank you for your encouraging comment!

---

### Review · Reviewer_ZAn5 · 2023-07-23

**Summary Of Contributions:**

This paper theoretically analyzes data transformation under the assumption that data are generated from a mixture of Gaussian distributions. In particular, the task  of clustering is considered as a representative application and a practical clustering algorithm is proposed. Under some appropriate assumptions, the authors show that perfect clustering is realized when the sample size goes to infinity. Empirical evaluation on synthetic and real-world datasets shows that the proposal is superior to existing clustering methods in term of the adjusted Rand index.

**Audience:**

Yes

**Broader Impact Concerns:**

I do not have any concerns.

**Claims And Evidence:**

Yes

**Requested Changes:**

1. Definition of $\mathcal{C}\_h$ in Eq.(6) is not clear to me. Now we are considering a single mixture distribution $\mathbb{P}\_{\mathbf{Z}}$, and a sample should come from the distribution. Thus, although $\mathbb{P}\_1$ and $\mathbb{P}\_2$ are components of $\mathbb{P}\_{\mathbf{Z}}$, the situation where a sample comes from either $\mathbb{P}\_1$ or $\mathbb{P}\_2$, which is considered in Eq.(6), does not make sense because this violates with the assumption of the mixture distribution with $0 < \pi\_1, \pi\_2 < 1$.
2. Theorems in Section 2.1 seem to be based on the assumption of Eq.(6). Hence I think what the authors are treating is clustering of a sample from not a mixture distribution but two independent distributions. So I am not sure why Theorem 2.2 can be used in the example in Section 2.1.1.
3. Around Eq.(6), $\mathbf{Z}\_1 ,\dots, \mathbf{Z}\_N$ are used as elements of a sample, while $\mathbf{Z}\_1$ and $\mathbf{Z}\_2$ appear and are used as random processes (variables) in the sentence immediately before Theorem 2.2 as the authors use the notation $\mathbb{P}\_{\mathbf{Z}\_1}$ and $\mathbb{P}\_{\mathbf{Z}\_2}$. Are these $\mathbf{Z}\_1$ and $\mathbf{Z}\_2$ two elements of the sample, or some difference processes? If so, what are they in this context of clustering?
4. Although the overview of the clustering process is given in Section 3.2, describing pseudo-code of the algorithm is also helpful for understanding.
5. Please describe the computational complexity of the proposed algorithm.
6. In Table 5.1, I guess $M$ means the sample size. Then please also write the number of features for each dataset.
7. In experiments, it seems that what "CD" in text and tables means is not explicitly explained. So please explain it.
8. All the real-world datasets in experiments are rather small scale. It would be interesting if larger scale, or higher-dimensional, datasets are also examined.

**Strengths And Weaknesses:**

## Strength
1. This paper is relevant as the problem of clustering and its theoretical analysis is still an important research topic.
2. This paper carefully analyzed the problem of clustering based on the Hajek and Feldman property. As the authors argue in Introduction, to date, this approach has not been studied at the sufficient depth, and the contribution of this paper is novel.
3. The proposed method shows good performance in experiments, which indicates that the contribution of this paper is not only theoretical but also potentially practical.

## Weaknesses
1. Overall this paper is hard to follow, and the presentation can be improved. One of the main reasons is that motivation and intuitive meaning has not been introduced when mathematical claims (Theorems) are stated. Thus it is difficult to understand why such theorems are required. In particular, in the context of clustering, how they are used in the task is often unclear.
2. As I state in the "requested changes" section, there are several concerns regarding the validity of the assumption in the theoretical analysis. Therefore, in its current form, the quality and the significance of theoretical analysis is not convincing.
3. Since experiments performed in this paper are not thorough, the quality of empirical evaluation is not high.

---

> ### Author Response · Authors · 2023-08-14
> **Revised version and response**
>
> Thank you for your constructive comments!
>
> We are now submitting a revised submission. All the changes in this revised version are now marked in Red color.
>
> A point by point response to the requested changes is in the file "RoR.pdf" (under Supplementary Material).

---

> > ### Comment · Reviewer_ZAn5 · 2023-08-28
> >
> > I am sorry for my late response.
> >
> > The quality of the manuscript is overall much improved. In particular, I understand the assumption of the theoretical analysis.
> > Nevertheless, I still have a few concerns:
> >
> > About my request 3, the authors say
> > > We have included a couple of sentences in page 5 (where Z1 and Z2 are selected) to explain clearly their roles.
> >
> > However, the important issue is that a "sample" and a "random variable" are mixed up, and $\mathbf{Z}_j$ refers to a sample in Eq. (6), while it refers to a random variable in Theorem 2 (and also other places), which should be resolved, e.g., using different alphabets.
> >
> > Also, about my request 7, the authors say
> > > The notation CD is related to the family names of the authors. To retain anonymity of our submission, we have continued to suppress its full form in the revised manuscript.
> >
> > but I do not understand it. It is totally fine to keep the anonymity, but the problem is that the reader does not have any idea about what "CD" is.
> > Do you mean that the authors' family names are used as the name of the proposal?
> > If so, it is enough to say something like "CD refers to our proposal" in the caption of the table without saying anything about its etymology.

---

> > > ### Author Response · Authors · 2023-08-28
> > > **Answer to your comments**
> > >
> > > Thank you very much for your new comments!
> > >
> > > Regarding your concerns, below we include our point of view on them.
> > >
> > > Concern 1:
> > >
> > > We apologize that we did not understand this concern. A sample is a collection of r.v.’s. Thus, in our meaning, the sentence before (6) can be written as:
> > >           Given a random sample composed by the r.v.’s $\Zvec_1,\ldots,\Zvec_N$ taken from …,
> > > or, in a shorter way:
> > >            Given $N$ i.i.d. r.v.’s $\Zvec_1,\ldots,\Zvec_N$ whose distribution is …
> > > So, we can rewrite the sentence in page 5 by saying:
> > >             Consider two random variables in a sample generated from …
> > >
> > > Concern 2:
> > >
> > > We agree that the meaning of CD is not obvious. But, once the names of the authors are disclosed, this confusion should get cleared. To be more precise, consider Table 4.1. In this table, the procedures CL (authors are Chiou and Li) and DHP (authors are Delaigle, Hall and Pham) appear. For completeness, we will add the line "CD refers to our proposal" in the text as you have suggested.

---

### Review · Reviewer_TJHS · 2023-07-26

**Summary Of Contributions:**

The paper presents some theoretical contributions relating to "perfect clustering" in the Gaussian Process (GP) mixture model framework for functional data.

The first set of results relate to the identifiability of the cluster "labels" of a fixed number of functions when viewed at finer and finer granularity via a sequence of associated finite dimensional distance metrics dependent on the covariance matrices of the projected mixture components, and the projection of the mixture itself. Because these distance functions depend only on the covariances their results do not directly allow for cluster identification in the scenario where only the mean functions of the components differ.
 The main result (from a clustering point of view) in this set states that the sequence of distances of the projections of two GPs tend in probability to distinct values depending on whether they are from two components with different covariance functions or not. Based on this they propose constructing a dissimilarity matrix in which the similarity between two processes is given as the average of the squared difference in distances between each of the two and each other "realisation" in the sample. Because of the previous result the dissimilarity will be zero (in probability) if they are from components with the same covariance and some non-zero value otherwise.

The second set of results relate to how this translates when the sequence of distance functions is based on sample covariance matrices in the increasing sample size context. The authors show that for (projection) dimension depending on the eigen-gaps of the sample covariance matrices they can essentially extend their distributional results extend to the sample context.

Some discussion on how to realise their results (not all of which are directly constructive) can be utilised in practice, along with experimental results on simulated and benchmark data show some promising practical implications as well.

**Audience:**

Yes

**Claims And Evidence:**

Yes

**Requested Changes:**

- The statement of Theorem 2.1 is strange to me. Since (4) does not in any way depend on the function b, it should come before. Something like: Let {A_d} be ... for which \lim_{d \to \infty} \frac {||\alpha_d||_\infty} {d} = 0. Then if b is such that (2) and (3) we get our convergence result."
- There is (as far as I can tell) a lack of consistency with when the covariance function (and its projections) is given its super-script Z (or Z_h depending on context) and when it is not. Am I missing some distinction here?
- There is also inconsistency in when the covariance function in the superscript of D^{\Sigma^Z_d}_d is given the subscript indicating it is actually a covariance function (otherwise it hasn't been defined when this supersctipt is not projected, even if there is no ambiguity in these cases).
- In (8) why do you have both notation L^{hk}_S and also L^{hk}?
- Statement (c) in Theorem 2.2 confuses me. In statements (a) and (b) you have had to assume the limits exist, whereas now suddenly these quantities suddenly exist and are also finite. Also, since h \not = k why is there not an L^k_S which would be distinct from L^h_S and L^{hk}_S? There is no "priority" of h over k?
- The definition of \hat u^Z_j above (17) seems never to be used. Why is it included?
- In the experiments, why is only the best result on the benchmark data sets given? All results for the simulations are given in the appendix and it would be interesting to know if the observation that the GMM applied on the \Gamma matrices continued to be the most consistent over kmeans and spectral clustering.

**Strengths And Weaknesses:**

Strengths:
- The results are persuasive, and at a high level the paper is relatively straightforward to follow even without too much deep technical knowledge.
- It is nice that what is a paper which (after some revision) could probably stand quite well only on its theoretical elements also contains some illustration for how it can be used practically, along with some reasonably satisfying performance results as well as code to reproduce the results

Weaknesses:
- My only major concern is that in trying to follow in detail, the notation could be made a lot clearer. This to the extent that it is not always clear that things are being correctly stated. I will detail some instances in the following section.

---

> ### Author Response · Authors · 2023-08-14
> **Revised version and response**
>
> Thank you for your constructive comments!
>
> We are now submitting a revised submission. All the changes in this revised version are now marked in Red color.
>
> A point by point response to the requested changes is in the file "RoR.pdf" (under Supplementary Material).

---

> > ### Comment · Reviewer_TJHS · 2023-08-26
> > **Thanks to the authors**
> >
> > Thank you to the authors for their thorough revision. I am overall happy with the changes, however I do still find some of the notation to be inconsistent. Having said that overall it has improved with the revision. I also think the paper would benefit tremendously from having it proof-read further, perhaps by an independent.

---

> > > ### Author Response · Authors · 2023-08-28
> > >
> > > Thank you for your encouraging comments! Once we have obtained feedback from all the reviewers and the Associate Editor, we will proofread our article once again to further improve the notation, and also request an independent researcher for feedback on our revised version.

---

### Decision · Action_Editors · 2023-09-05

**Recommendation:** Accept with minor revision

**Comment:**

All three reviewers are leaning accept on the paper.  The revision in particular helped to clarify some of the concerns.  One reviewer still feels that the presentation of the paper could improve, and I would recommend doing a further pass to improve readability of the paper.   In particular, that reviewer stated "the paper would benefit from being proofread by a native English speaker as the grammar is at times awkward to read, not to mention sometimes incorrect."  I think in general the paper just needs a polishing pass---nothing specific has been noted by the reviewers but try to smooth it out. Otherwise, this looks like it's in good shape.

**Audience:**

The problem area is definitely relevant to the readers of TMLR.  In particular, clustering is a fundamental problem in ML.

**Claims And Evidence:**

The reviewers are in agreement here that the results are persuasive, and contain both theoretical insights as well as solid empirical results.  The authors in particular present some novel theoretical results on clustering in the GP mixture model framework (sections 2 and 3, including results on convergence and consistency), and show empirical results comparing to relevant baselines in sections 4 and 5.  A revision was submitted which clarified many/most of the concerns of the reviewers on some of the claims made in the paper.

---

> ### Author Response · Authors · 2023-09-14
> **Many thanks!**
>
> As authors, we would like to offer our sincere thanks to the AE and the reviewers for their time and effort in reviewing the paper. We are grateful for the feedback which has improved the quality of the paper.
>
> We have now uploaded a camera-ready version of our paper.